# Global Geometry Is Not Enough for Vision Representations

**Jiwan Chung** [1]  **Seon Joo Kim** [1]

## Abstract

A common assumption in representation learning is that globally well-distributed embeddings support robust and generalizable representations. This focus has shaped both training objectives and evaluation protocols, implicitly treating global geometry as a proxy for representational competence. While global geometry effectively encodes which elements are present, it is often insensitive to how they are composed. We investigate this limitation by testing the ability of geometric metrics to predict compositional binding across 19 vision encoders. We find that standard geometry-based statistics exhibit near-zero correlation with compositional binding. In contrast, functional sensitivity, as measured by the input–output Jacobian, reliably tracks this capability. We further provide an analytic account showing that this disparity arises from objective design, as existing losses explicitly constrain embedding geometry but leave the local input–output mapping unconstrained. These results suggest that global embedding geometry captures only a partial view of representational competence and establish functional sensitivity as a critical complementary axis for modeling composite structure.

## 1. Introduction

A central objective of visual representation learning is to produce embeddings that support stable and transferable downstream behavior. A dominant view underlying recent progress is geometric, namely that representation quality follows from enforcing global regularity of the embedding distribution, such as avoiding collapse, anisotropy, or redundancy (Wang & Isola, 2020; Ethayarajh, 2019), as illustrated in Figure 2a. This view motivates many modern self-supervised objectives, including contrastive alignment

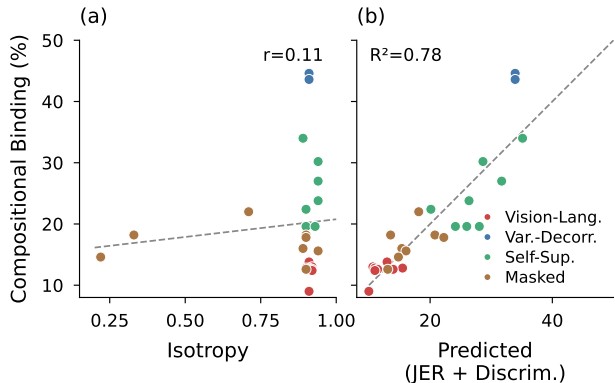

*Figure 1.* **Global geometry is not enough for composition modeling. (a)** Global isotropy does not strongly correlate with compositional binding across different types of vision encoders. **(b)** In contrast, our functional metric (Jacobian Effective Rank + Discriminability) accurately predicts performance. Statistics in Table 2.

methods such as CLIP (Radford et al., 2021), prototype-based self-distillation approaches such as DINO (Caron et al., 2021), and redundancy-reduction formulations such as VICReg (Bardes et al., 2022a) and Barlow Twins (Zbontar et al., 2021), and is supported empirically on standard downstream tasks driven by feature presence, such as linear-probe classification and image-text retrieval.

This geometric view has recently been further formalized through explicit theoretical claims about optimal representation geometry. In particular, LeJEPA (Balestriero & LeCun, 2025) identifies the isotropic Gaussian as an optimal embedding distribution for minimizing downstream prediction risk under the JEPA framework and its assumptions, and introduces a regularizer to enforce this structure. In parallel, representation quality is commonly evaluated using static, distributional metrics that summarize global embedding geometry, such as isotropy or covariance spectra (Ethayarajh, 2019; Wang & Isola, 2020).

At the same time, visual scenes are defined not only by which elements are present, but by how those elements are composed. The problem of compositional binding has been studied in cognitive science for decades (Treisman & Gelade, 1980; Roskies, 1999). Scenes can share identical feature statistics, such as shapes, colors, and parts, yet differ fundamentally in their arrangement. For example, placing

[1]Yonsei University. Correspondence to: Jiwan Chung <jiwan.chung.research@gmail.com>, Seon Joo Kim <seonjookim@yonsei.ac.kr>.

*Proceedings of the 43rd International Conference on Machine Learning*, Seoul, South Korea. PMLR 306, 2026. Copyright 2026 by the author(s).

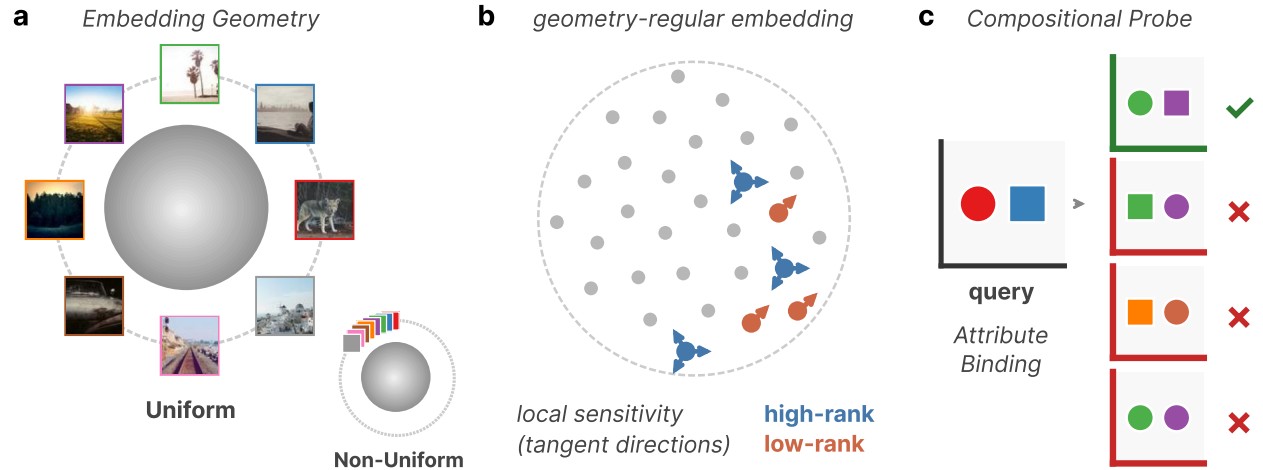

*Figure 2.* **(a)** Existing visual representation learning objectives and metrics often assume that favorable geometric properties, such as *uniformity*, yield optimal representations. **(b)** However, we show that geometry is not everything, as *functional sensitivity* can differ between geometrically well-distributed representations. **(c)** To examine the downstream implications of this difference, we present the synthetic attribute-binding evaluation setting used in this work, designed to isolate confounding variables such as object detection.

a washer on top of a nut rather than beneath it preserves the parts inventory but breaks the assembly. Without sensitivity to such structure, a representation collapses to a *bag-of-features*, encoding what is present but not how it is bound (Csurka et al., 2004; Brendel & Bethge, 2019; Geirhos et al., 2019).

In this work, we analyze representations learned under global geometric regularity, often assumed to encode which elements are present (Chen et al., 2020; Grill et al., 2020; Assran et al., 2023), and show that they do *not* encode how those elements are composed (Section 3). To establish this, we design synthetic probes for cleanly isolating compositional binding capacity from confounding factors such as natural object detection. We then perform a comprehensive evaluation on various visual encoders spanning self-supervised, supervised, and vision–language families. Across models, we find that standard geometric metrics exhibit near-zero correlation with binding probe accuracy, indicating that geometric regularity and compositional structure are statistically orthogonal properties (Figure 1).

However, we find that *functional sensitivity* exhibits a strong correlation with concept binding performance (Section 4). Functional sensitivity characterizes how an encoder responds to structured variations in its input at the level of the input-output mapping, rather than through aggregate properties of the embedding distribution. We measure this property using the effective rank of the input-output Jacobian (Roy & Vetterli, 2007; Pennington et al., 2017), which summarizes the dimensionality of local variations in representation space induced by input perturbations.

Further, we examine whether alternative geometry-based readouts can access compositional binding from frozen rep-

resentations (Section 5). Across models, neither kNN nor local PCA improves binding accuracy over cosine similarity, and both often degrade performance, including for the strongest encoders. This indicates that binding information is neither concentrated in local neighborhoods nor aligned with dominant variance directions. Qualitative analysis shows that near-chance models assign high similarity to both targets and distractors but fail to produce margins reflecting correct shape–position bindings. Together, these results indicate that linear accessibility of compositional structure is intrinsic to the pretraining objective and cannot be recovered through alternative geometric readouts.

Crucially, most existing visual encoder objectives emphasize global geometric regularities in the representation, while leaving the structure of local functional sensitivity largely underspecified. As detailed in Section 6, standard losses optimize distribution-level statistics (e.g., similarity or reconstruction error) and therefore constrain the input-output mapping only through indirect aggregation over perturbations, rather than explicitly shaping its directional sensitivity. Variance–decorrelation objectives such as Barlow Twins and VICReg differ in that they penalize redundant variation across representation dimensions.

Our contributions:

- *Limits of geometry.* We demonstrate that widely used geometric diagnostics reliably track performance on tasks driven by single-factor variation, yet systematically fail on tasks requiring genuine compositional structure.

- *Functional rank as a compositional diagnostic.* We introduce Jacobian effective rank as a functional measure of local input-output sensitivity, and show that it predicts com-

*Table 1.* **Structural inductive biases of visual encoders.** Models grouped by training objective and induced geometric properties.

| Family | Examples | Bias | Geometric tendency |
| --- | --- | --- | --- |
| **Variance Decorrelation** | Barlow Twins, VICReg | Redundancy reduction | High isotropy and participation ratio (whitened eigenvalue spectrum) |
| **Masked modeling** | MAE, BEiT, BEiTv2 | Context prediction | Embedding stability under masking (low sensitivity to structured occlusion) |
| **Clustering / distill.** | DINO, DINOv2, SwAV | Assignment consistency | Low intra-cluster variance with clear inter-cluster separation |
| **Contrastive** | MoCo v3 | Instance discrimination | Uniform hyperspherical spread (high uniformity, low anisotropy) |
| **Vision-language** | CLIP, SigLIP, EVA-CLIP | Cross-modal alignment | Semantically aligned geometry with strong linear separability |
| **Supervised** | ConvNeXt, ViT | Label prediction | Margin-based class separation in embedding space |

positional capacity across a broad set of encoders when measured via a standardized Jacobian-spectrum probe.

- *Objective-induced sensitivity structure.* We show that the differences arise predictably from the training objective: objectives that preserve or decorrelate local sensitivity support compositional structure, whereas others bias representations toward single-factor discrimination.

**Conflict of Interest Disclosure.**  The authors declare no financial conflicts of interest.

## 2. Related Work

**Geometric Evaluation of Representations.**  A large body of recent work evaluates representation quality through the global geometry of embedding distributions, using alignment, uniformity, isotropy, rank, and variance as primary diagnostics (Wang & Isola, 2020; Papyan et al., 2020). Within this paradigm, well-conditioned geometry is treated as a proxy for expressivity and information preservation, and is repeatedly linked to improved transfer and robustness across tasks (Huang et al., 2024; Garrido et al., 2023).

Alignment and uniformity provide a canonical geometric decomposition of contrastive objectives and remain a common lens for analyzing training dynamics and collapse avoidance (Saunshi et al., 2019; Cabannes et al., 2023). Subsequent work often frames representational collapse as a geometric failure mode, characterized by loss of rank, variance, or effective dimensionality, including in large-scale and non-contrastive settings (Papyan et al., 2020; Li et al., 2022; He et al., 2024; Sansone et al., 2025).

Isotropy has emerged as a complementary diagnostic, typically quantified via the covariance spectrum of embeddings (Ethayarajh, 2019). Empirical studies report strong anisotropy in learned representations and motivate normalization or regularization strategies that explicitly target

isotropy and rank preservation (Mu & Viswanath, 2018; Gao et al., 2021; Su et al., 2021). Recent theoretical analyses further consolidate this geometric emphasis by characterizing representation quality in terms of alignment, variance, and conditioning, and by identifying isotropic representations as risk-optimal under predictive objectives (Assran et al., 2023; Balestriero & LeCun, 2025).

A complementary class of geometric summaries operates on pairwise *inter-model* representational distances rather than single-model statistics. Linear CKA (Kornblith et al., 2019), Procrustes shape distance, and related measures formalized as proper metric spaces on neural representations (Williams et al., 2021) compare response manifolds across models on shared stimuli. Recent work (Bo et al., 2024; Braun et al., 2025) reports that inter-model shape metrics align with general functional differences between models, while emphasizing that functional and representational similarity can dissociate. In Section D, we test whether these inter-model metrics predict compositional binding; they do not.

**Objective-Driven Geometric Regularization.**  This geometric perspective is directly reflected in the design and interpretation of dominant representation learning objectives. Contrastive methods promote uniformity through negative sampling, inducing global dispersion in the embedding space while preserving alignment of positive pairs (Chen et al., 2020; He et al., 2020). Non-contrastive objectives pursue the same geometric goals more explicitly, preventing collapse through variance preservation and decorrelation and thereby encouraging high-rank, approximately isotropic representations (Grill et al., 2020; Chen & He, 2021; Zbontar et al., 2021; Bardes et al., 2022a;b).

Similar geometric analyses extend to multimodal representation learning. Vision-language models are commonly studied through the structure of a shared embedding space, with alignment, anisotropy, and large-scale geometric organiza-

*Table 2.* **Correlation of metrics with attribute binding accuracy (Pearson** $r$**).** $p$ denotes $p$-value. $R^2$ in parentheses denotes leave-one-out cross-validated.

| Category | Metric | $r$ | $p$ | $R^2$ |
|---|---|---|---|---|
| Geometry | G.PR | +0.10 | 0.640 | - |
| | G.Iso | +0.11 | 0.593 | - |
| | L.Iso | +0.02 | 0.933 | - |
| Functional | JER | +0.69 | 0.0001 | 0.47 |
| | JER$_{null}$ | -0.16 | 0.451 | - |
| | $\Delta$JER | +0.68 | 0.0001 | 0.47 |
| | JER + Disc. | - | - | 0.78 (0.71) |

tion analyzed in terms of factors for performance and failure modes (Radford et al., 2021; Cherti et al., 2023; Levi & Gilboa, 2025; Eslami & de Melo, 2025). Recent work further proposes objective-level or architectural modifications to address geometric pathologies observed in such joint spaces (Kang et al., 2025).

**Limits of Static Geometric Evaluation.** Despite differences in objective design and modality, these approaches share a common evaluative practice: representation quality is primarily assessed through static geometric summaries of embedding distributions (Jing et al., 2022). Such summaries abstract away the functional dependence of representations on structured input variation.

Recent empirical studies indicate that strong geometric properties do not guarantee compositional or relational behavior. In particular, vision-language models with well-aligned and transferable embeddings have been shown to behave like bags of words, exhibiting insensitivity to word order and compositional structure (Yuksekgonul et al., 2023). Related work on compositional generalization reports systematic failures on tasks requiring structured recombination, despite representations that appear well-formed under standard geometric criteria (Hua et al., 2024; Kamath et al., 2024; Zheng et al., 2024; Camposampiero et al., 2025). These findings motivate evaluation frameworks that move beyond static geometry toward analyses of functional sensitivity.

## 3. The Limit of Geometry

We test the hypothesis that geometric regularity predicts compositional capability by evaluating 26 vision encoders of different training objectives on synthetic binding tasks (Section 3.1). Across all models, we observe no significant correlation between standard geometric metrics and binding performance (Section 3.2). This null result remains robust across metric definitions and hyperparameter choices.

### 3.1. Experimental Setup

Our synthetic task is *intentionally designed to be as minimal as possible*, eliminating nuisance factors to isolate the model's intrinsic capacity for compositional representation.

**Models.** We evaluate 26 pretrained vision encoders spanning seven distinct objective families (Table 1): contrastive (MoCo v3), variance-decorrelation (Barlow Twins, VICReg), non-contrastive clustering (SwAV), self-distillation (DINO, DINOv2), masked prediction (MAE, BEiT, BEiTv2), vision-language (CLIP, SigLIP, EVA-CLIP), and supervised baselines (ConvNeXt, ViT). This suite covers multiple scales (ViT-S/B/L, ResNet-50) and architectures to disentangle objective-specific properties from model size or inductive bias. All models are evaluated in inference mode using standard ImageNet preprocessing.

**Target Task: Compositional Binding.** To isolate compositional capability from texture recognition, we employ the *Attribute Binding* benchmark (Figure 2c). Generated via the protocol detailed in Appendix H, this task utilizes geometric primitives to test relational assignment. The model must match a query to a target based on shape-position bindings. Crucially, the query and target use *completely disjoint color sets* (e.g., Query: {Red, Green} vs. Target: {Blue, Yellow}). This prevents any reliance on color co-occurrence, forcing the model to represent abstract spatial relations. *The task deliberately avoids visual complexity, texture variation, and semantic cues.* This minimal design ensures that failures cannot be attributed to perceptual difficulty, but instead reflect limitations in compositional binding itself.

**Structural Control: Same/Different (Disc.).** Even under this deliberately simplified setting, binding performance may still be affected by factors unrelated to compositional assignment, such as incomplete invariance to appearance. To characterize these effects, we employ a *Structural Discrimination* probe. This is a binary classification task in which the model must determine whether two images share the same *spatial configuration* (e.g., "circle left of square") despite differing in color. Because the task isolates structural discrimination under appearance variation without requiring relational matching across disjoint attribute sets, it helps interpret variation in binding performance that persists even in the minimal synthetic regime. A model may pass Same/Different (detect that two configurations differ) yet fail Binding (select the configuration that matches a query against alternatives sharing the same parts); for example, DINOv2 ViT-B/14 scores 74.6% on Same/Different but only 30.2% on Binding. Refer to Appendix J for further detail.

**Geometric Metrics.** We compute distributional statistics on the ImageNet-1k validation set. Let $\lambda_1 \geq \cdots \geq \lambda_D$ be

*Table 3.* **Evaluation results across vision encoders** sorted by Attribute Binding accuracy. Geometric metrics show no pattern with binding. JER measures functional sensitivity; Same/Diff probes basic discriminability.

| Model | Arch | Objective | Geometry | | | Functional Probes | | Task |
| | | | G.PR | G.Iso | L.Iso | **JER** | Disc. | Binding |
|---|---|---|---|---|---|---|---|---|
| Barlow Twins | ResNet-50 | Var-Decorr | 0.11 | 0.91 | 0.82 | **28.8** | 75.0 | 44.6 |
| VICReg | ResNet-50 | Var-Decorr | 0.06 | 0.91 | 0.82 | **28.8** | 75.0 | 43.6 |
| SwAV | ResNet-50 | Clustering | 0.08 | 0.89 | 0.80 | **29.9** | 75.0 | 34.0 |
| DINOv2 | ViT-B/14 | Self-Distill | 0.22 | 0.94 | 0.82 | **24.2** | 74.6 | 30.2 |
| DINOv2 | ViT-S/14 | Self-Distill | 0.29 | 0.94 | 0.82 | **26.8** | 75.0 | 27.0 |
| DINOv2 | ViT-L/14 | Self-Distill | 0.21 | 0.94 | 0.83 | **23.1** | 72.4 | 23.8 |
| MoCo v3 | ViT-B/16 | Contrastive | 0.11 | 0.90 | 0.85 | **16.3** | 75.0 | 22.4 |
| MAE | ViT-L/16 | Masked | 0.01 | 0.71 | 0.72 | **25.5** | 50.0 | 22.0 |
| DINO | ViT-S/16 | Self-Distill | 0.21 | 0.90 | 0.81 | **26.3** | 68.6 | 19.6 |
| DINO | ViT-B/16 | Self-Distill | 0.11 | 0.90 | 0.83 | **24.0** | 69.6 | 19.6 |
| DINOv2 | ViT-g/14 | Self-Distill | 0.14 | 0.93 | 0.82 | **21.7** | 71.0 | 19.6 |
| MAE | ViT-B/16 | Masked | 0.00 | 0.33 | 0.56 | **27.9** | 50.0 | 18.2 |
| ViT | ViT-L/16 | Supervised | 0.11 | 0.90 | 0.80 | **14.9** | 64.6 | 18.2 |
| BEiTv2 | ViT-B/16 | Masked | 0.11 | 0.90 | 0.79 | **19.8** | 71.4 | 17.8 |
| ConvNeXt | Large | Supervised | 0.12 | 0.89 | 0.79 | **14.1** | 70.2 | 16.0 |
| ViT | ViT-B/16 | Supervised | 0.19 | 0.94 | 0.79 | **17.1** | 64.8 | 15.6 |
| ConvNeXt | Base | Supervised | 0.00 | 0.22 | 0.79 | **18.7** | 58.6 | 14.6 |
| CLIP | ViT-B/16 | Vision-Lang | 0.14 | 0.91 | 0.85 | **20.8** | 50.0 | 13.8 |
| CLIP | ViT-L/14 | Vision-Lang | 0.13 | 0.92 | 0.86 | **18.7** | 50.0 | 13.0 |
| SigLIP | ViT-B/16 | Vision-Lang | 0.09 | 0.91 | 0.83 | **23.1** | 50.0 | 12.8 |
| SigLIP | SoViT-400M | Vision-Lang | 0.07 | 0.90 | 0.82 | **19.0** | 50.0 | 12.8 |
| BEiT | ViT-B/16 | Masked | 0.07 | 0.90 | 0.72 | **20.9** | 50.0 | 12.6 |
| CLIP | ViT-B/32 | Vision-Lang | 0.14 | 0.91 | 0.85 | **21.7** | 50.0 | 12.6 |
| EVA-CLIP | ViT-E/14 | Vision-Lang | 0.08 | 0.92 | 0.85 | **19.4** | 50.0 | 12.6 |
| EVA-CLIP | ViT-B/16 | Vision-Lang | 0.14 | 0.92 | 0.84 | **19.0** | 50.0 | 12.4 |
| EVA-CLIP | ViT-L/14 | Vision-Lang | 0.10 | 0.91 | 0.83 | **18.1** | 50.0 | 9.0 |

the eigenvalues of the embedding covariance matrix. We measure:

- *Global Participation Ratio (G.PR):* $(\sum \lambda_i)^2 / (D \cdot \sum \lambda_i^2)$, the normalized effective dimensionality of the global manifold (Rudelson & Vershynin, 2007).

- *Global Isotropy Score (G.Iso):* $1 - \lambda_1 / \sum \lambda_i$, measuring how much variance is distributed beyond the top principal component (Ethayarajh, 2019; Mu & Viswanath, 2018).

- *Local Isotropy (L.Iso):* For each sample, we compute the covariance matrix of its $k$-nearest neighbors in embedding space, then measure Isotropy Score on this local covariance. The reported value is the average across 500 randomly sampled points. This serves as a proxy for Local Intrinsic Dimensionality (Levina & Bickel, 2004; Ma et al., 2018). We use $k = 32$ in the main analysis.

### 3.2. Results

The results are summarized in Table 3, with correlation statistics reported in Table 2. Across the evaluated models, we observe no meaningful association between geometric regularity metrics and compositional binding perfor-

mance. *Global Participation Ratio* ($r = 0.10, p = 0.64$) and *Isotropy Score* ($r = 0.11, p = 0.59$) fail to predict binding accuracy, indicating that the global effective dimensionality is unrelated to its structural composability.

Similarly, *Local Isotropy Score* exhibits a weak and statistically insignificant correlation ($r = 0.02, p = 0.93$). While one might hypothesize that locally isotropic manifolds facilitate better separation of compositional states, the data suggests that high local dimensionality is neither necessary nor sufficient for binding.

These findings challenge the prevailing assumption that optimizing for geometric properties such as uniformity or isotropy intrinsically improves the representation of compositional structure. Our results indicate that geometric regularity is effectively orthogonal to compositional capability; high-quality geometric metrics do not translate to functional binding performance.

**Robustness Analysis.** To ensure the null result for Local Isotropy is not an artifact of hyperparameter or metric choice, we conduct two ablations. First, we vary the neighborhood size across $k \in \{8, 16, 32\}$; no setting yields a significant correlation (all $p > 0.8$), as shown in Table 4. Second, we test alternative formulations of local geometry: Effective Rank (the exponential of the eigenvalue entropy)

*Table 4.* Robustness of null result for local isotropy. Neither neighborhood size ($k$) nor metric formulation yields significant correlation with binding (Pearson $r$).

| Setting | $r$ | $p$-value |
|---|---|---|
| *Neighborhood size ($k$)* | | |
| k=8 | +0.045 | 0.829 |
| k=16 | +0.020 | 0.924 |
| k=32 | +0.017 | 0.933 |
| *Metric formulation (at k=16)* | | |
| Variance $(1 - \lambda_1/\Sigma\lambda)$ | +0.020 | 0.924 |
| Effective Rank | -0.031 | 0.881 |
| Participation Ratio | -0.031 | 0.881 |

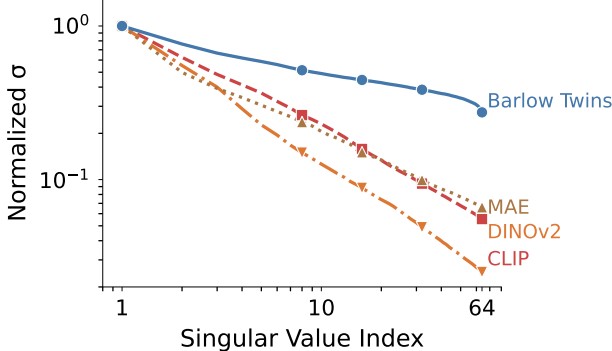

*Figure 3.* **Normalized singular value spectrum of the input-output Jacobian for four models.** Barlow Twins maintains a flat spectrum indicating uniform sensitivity across input directions. CLIP, DINOv2, and MAE show rapid spectral decay, concentrating sensitivity in few directions.

and the unnormalized Participation Ratio. All metrics fail equally ($p > 0.8$). These results confirm that the dissociation between local geometric regularity and compositional binding is robust to both the scale and the functional form of the local manifold analysis.

## 4. Functional Sensitivity

Having established that static geometric properties fail to predict compositional capability, we investigate a functional alternative: the encoder's sensitivity to input perturbations. We hypothesize that models capable of binding must maintain a high effective dimensionality in their input-output mapping, utilizing many independent directions to encode variations. We test this by estimating the effective rank of the Jacobian matrix (Section 4.1) and evaluating its predictive power against the binding benchmarks (Section 4.2).

### 4.1. Jacobian Effective Rank

We characterize the *functional sensitivity* of an encoder $f : \mathbb{R}^M \to \mathbb{R}^D$ through its input-output Jacobian $J_f(x) = \frac{\partial f(x)}{\partial x} \in \mathbb{R}^{D \times M}$. The singular values $\{\sigma_i(x)\}$ of $J_f(x)$ quantify the strength of local input-output sensitivity: larger $\sigma_i$ correspond to input directions that induce larger first-order changes in the representation, and their distribution reflects how many independent directions the model is locally sensitive to.

We evaluate Jacobians on natural images from the ImageNet validation set, processed with standard ImageNet normalization. Section E.1 verifies that the same finding holds when the Jacobian is instead evaluated on Gaussian noise ($x \sim \mathcal{N}(0.45, 0.225^2)$, a scalar approximation of ImageNet per-channel pixel statistics, clipped to $[0, 1]$), a stimulus-agnostic probe of local conditioning decoupled from natural-image semantics. Full hyperparameters are provided in Appendix I (Table 14); wall-clock cost is reported in Section F. Since explicitly forming $J_f(x)$ is intractable at image resolution, we estimate its leading singular values using a randomized range-finding procedure (Halko et al.,

2011), computing Jacobian-vector products for $k$ random orthonormal probe directions via automatic differentiation.

Given the estimated singular values $\{\sigma_i\}_{i=1}^k$, we define the *Jacobian Effective Rank (JER)* using the same *Participation Ratio (PR)* functional form applied to the Jacobian spectrum,

$$\text{JER}(x) = \frac{\left(\sum_{i=1}^k \sigma_i\right)^2}{\sum_{i=1}^k \sigma_i^2}, \qquad \text{JER} = \mathbb{E}_x[\text{JER}(x)]. \quad (1)$$

While this uses the identical PR formula as the Global Participation Ratio, the quantity being measured is fundamentally different. G.PR is computed from the covariance eigenvalues of embeddings across a dataset and characterizes *distributional embedding geometry*, whereas JER applies PR to the singular values of the input-output Jacobian and directly characterizes *local functional sensitivity* of the encoder.

### 4.2. Results

Results are reported in Table 3, with correlation statistics in Table 2. In stark contrast to the geometric metrics, the Jacobian Effective Rank on natural images is a strong predictor of compositional binding. We observe a positive correlation ($r = 0.69, p < 0.001$) between the effective rank and performance on the Attribute Binding task. This suggests that models maintaining a higher effective dimensionality in their input-output mapping possess the necessary functional capacity to bind disparate features.

To rule out architectural baselines, we computed JER on the Gaussian-noise probe (Section E.1) for both the trained and randomly-initialized versions of all 26 models. JER on random initialization is uncorrelated with binding ($r = -0.16$, $p = 0.45$) and clusters tightly across architectures ($30.86 \pm 1.93$), while JER on trained models varies widely ($18.41 \pm 5.44$). The training-induced change $\Delta\text{JER} = \text{JER}_{\text{trained}} - \text{JER}_{\text{null}}$ is negative for every model

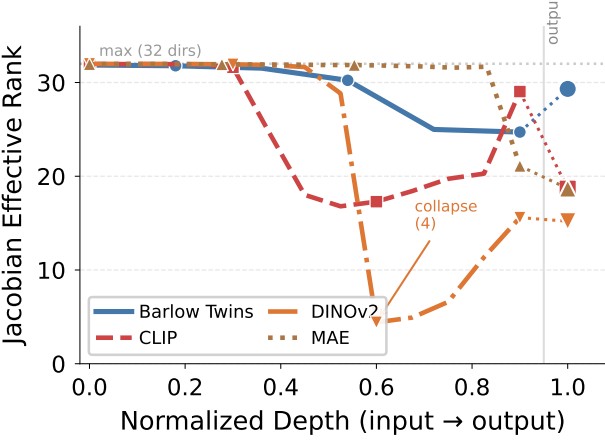

*Figure 4.* **Evolution of Jacobian effective rank across network depth.** Effective rank is computed at each block using random noise inputs. Barlow Twins preserves high rank throughout the backbone. CLIP and DINOv2 exhibit mid-network rank collapse (blocks 5–8). MAE maintains high rank in the backbone but collapses at the projection layer.

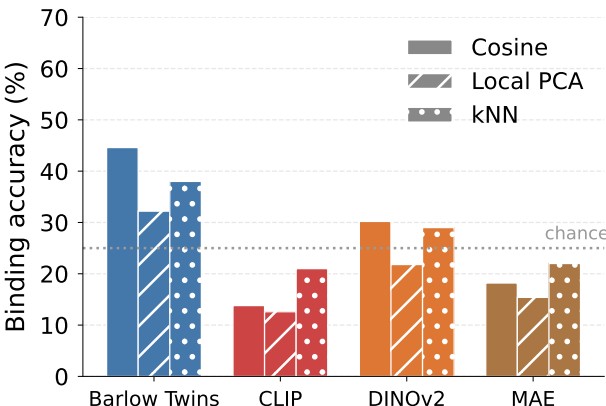

*Figure 5.* **Geometry-based readout comparison results.** Zero-shot cosine similarity, local PCA, and kNN are evaluated on frozen embeddings. Barlow Twins and VICReg perform well with cosine similarity, while CLIP, DINOv2, and MAE remain near chance under all geometric readouts.

(mean $-12.46$): training systematically reduces functional rank from its initialization baseline.

The hierarchy of models reveals a clear trend: variance-decorrelation objectives (e.g., Barlow Twins, VICReg) consistently exhibit the highest effective ranks ($\sim 28.8$) and the strongest binding performance ($\sim 44\%$). Conversely, contrastive vision-language models (e.g., CLIP) show collapsed rank ($\sim 20$) and near-chance binding accuracy ($13\%$).

Furthermore, we find that functional sensitivity and structural discrimination are complementary predictors. A bivariate regression model combining Jacobian Effective Rank with the Same/Different accuracy explains $78\%$ of the variance in binding performance ($R^2 = 0.78$; LOO-CV $R^2 = 0.71$). This indicates that compositional capability depends on both the intrinsic complexity of the encoder (measured by Jacobian rank) and its specific invariance properties (measured by Same/Different). Confound controls and seed stability are reported in Appendix A.3 and Appendix A.4.

### 4.3. Mechanisms of Functional Rank Collapse

The final-layer JER values in Table 3 are the endpoint of layer-wise spectral trajectories that differ qualitatively across objective families. We characterize these trajectories through the full singular-value spectrum at the encoder output (Figure 3) and the depth-resolved effective rank (Figure 4); together they reveal *where* the rank gap observed at the output originates.

The Jacobian singular-value spectra are shown in Figure 3. Variance-decorrelation models such as Barlow Twins exhibit

a comparatively flat, slowly decaying spectrum, indicating uniform sensitivity across many input directions. In contrast, CLIP, DINOv2, and MAE display rapid spectral decay, concentrating functional sensitivity in a small number of dominant modes. This collapse explains the reduced effective rank observed for these models.

To localize the source of this functional collapse, we track the Jacobian Effective Rank across network depth (Figure 4). Barlow Twins preserves a high effective rank throughout the backbone, indicating distributed sensitivity at all stages of processing. In contrast, CLIP and DINOv2 exhibit a pronounced mid-network collapse (blocks 5–8), with DINOv2 briefly dropping to rank 4 before partial recovery. MAE maintains high rank through the backbone but collapses sharply at the projection layer, suggesting that its functional bottleneck is introduced only at the output interface.

## 5. Further Analysis

We provide additional analysis to characterize how the JER signal transfers and how binding structure manifests in pretrained representations. We first validate JER on a natural-image compositional benchmark (Section 5.1). We then examine whether alternative geometry-based readouts can access binding information from frozen embeddings (Section 5.2), and qualitatively analyze observed failure cases (Section 5.3).

### 5.1. Natural-Image Validation

To assess whether the synthetic-benchmark finding transfers to natural images, we evaluate JER's predictive power on SugarCrepe (Hsieh et al., 2023), a benchmark of 7,511 natural-image compositional trials spanning seven attribute

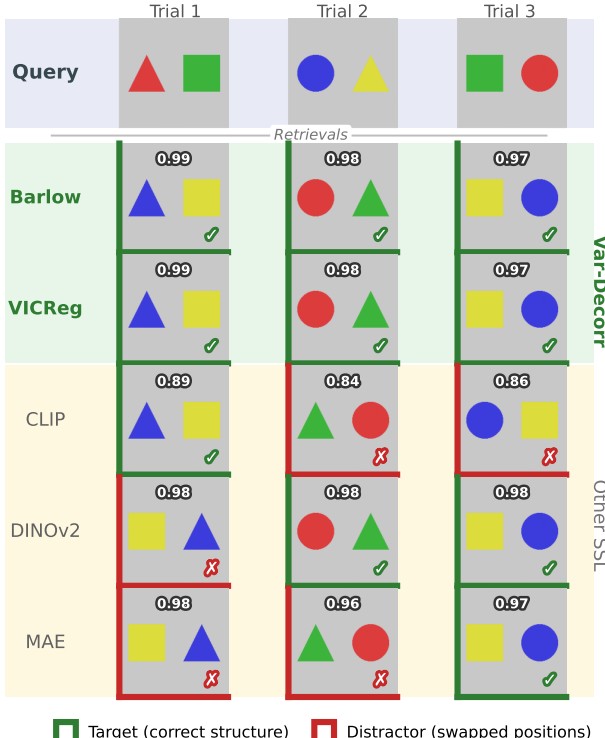

*Figure 6.* **Qualitative examples of binding success and failure.** Queries and targets share no colors, requiring positional binding. Barlow Twins and VICReg retrieve the correct target, whereas CLIP, DINOv2, and MAE often select structurally incorrect distractors despite high similarity.

and object subtypes (additions, replacements, swaps). We compare 13 CLIP variants that share the contrastive image-text objective but differ in training data (WebImageText, LAION-2B, DataComp, MetaCLIP, DFN) and scale (ViT-B/32, B/16, L/14); self-supervised encoders are excluded because SugarCrepe requires text-image alignment. The other vision-language encoders in our suite, EVA-CLIP and SigLIP, are also excluded from this comparison: they differ in architecture and training objective, which would confound a comparison designed to isolate the effect of training data and scale; they are evaluated on the main binding benchmark (Table 3).

JER predicts SugarCrepe performance with $r = 0.69$, $p = 0.009$ (Pearson, on the canonical sample-weighted over-all score). The relationship is robust to architecture: within the four ViT-L/14 variants alone, JER predicts SugarCrepe nearly perfectly ($r = 0.97$, $p = 0.029$). After controlling for ImageNet zero-shot accuracy and architecture, JER retains its predictive signal while ImageNet accuracy does not (Section B). The synthetic-benchmark finding therefore generalizes to natural-image compositional reasoning, within the CLIP-family scope.

## 5.2. Geometry-Based Readouts

Figure 5 compares the performance of different geometry-based readouts across representative models. For CLIP, DINOv2, and MAE, all three methods (i.e., cosine similarity, kNN, and local PCA) perform near or below chance level (25%). This indicates that the relative layout of embeddings does not reliably reflect compositional binding for these models.

In contrast, variance-decorrelation models (Barlow Twins and VICReg) achieve strong performance using cosine similarity alone ($\sim$44%). Importantly, neither kNN nor local PCA improves performance for any model; both interventions consistently degrade accuracy, including for the highest-performing encoders. This suggests that binding information is neither concentrated in local neighborhoods nor recoverable by projecting onto dominant variance directions.

Overall, these results show that purely geometric post-hoc readouts are insufficient to extract compositional structure from most pretrained representations. Linear accessibility of binding appears to be an intrinsic property induced by the pretraining objective, rather than something that can be recovered through alternative distance metrics or local manifold operations.

## 5.3. Qualitative Study

We further perform a qualitative analysis (Figure 6). Each trial is constructed such that the query and correct target share no colors, forcing the model to rely exclusively on structural binding (i.e., which shape appears in which position) rather than feature overlap.

Variance-decorrelation models (Barlow Twins and VICReg) consistently retrieve the correct target across all trials, achieving perfect accuracy with stable positive similarity margins. In contrast, CLIP, DINOv2, and MAE frequently select distractors that preserve object identity but violate positional binding, resulting in near-chance performance.

Crucially, these failures are not explained by low similarity scores. DINOv2 and MAE assign extremely high cosine similarity to both targets and distractors (often exceeding 0.97), yet exhibit negligible or negative margins between them. This indicates that while these models recognize the images as globally similar, they fail to discriminate *how* they are similar. High embedding similarity therefore does not imply correct compositional understanding.

## 6. Objective–Sensitivity Alignment

Our empirical results show that Jacobian Effective Rank (JER), a statistic of the encoder Jacobian, strongly correlates with downstream performance on tasks that go beyond

bag-of-features similarity, such as attribute binding. We argue that this relationship is mechanistic: the training objective affects which directions of functional sensitivity are preserved in the learned representation.

Here, we provide a simplified theoretical account of how different objectives constrain the encoder Jacobian through the functionals they optimize. Each objective exposes only a limited projection of $J(x)$ to optimization, shaping the geometry of the learned Jacobian according to what the loss can observe. For clarity, we keep the analysis intuitive and defer full derivations to the appendix (Appendix G).

### 6.1. Local Sensitivity under Augmentation

Let $f : \mathbb{R}^n \to \mathbb{R}^d$ be an encoder and $z = f(x)$. Consider an augmented view $x' = x + \delta$ with $\mathbb{E}[\delta] = 0$ and covariance $\Sigma_\delta$. A first-order expansion gives

$$z' \approx z + J(x)\delta, \qquad (2)$$

where $J(x) = \partial f/\partial x$ is the encoder Jacobian. Conditioned on $x$, the feature covariance from augmentation satisfies

$$\mathrm{Cov}_\delta(z' \mid x) \approx J(x)\Sigma_\delta J(x)^\top. \qquad (3)$$

Equation (3) provides the basic link between the objective and the local input-output sensitivity of the encoder.

### 6.2. Which Functionals Are Constrained by the Objective?

Let $\mathcal{L}$ denote a training objective. To first order, optimization constrains only those components of $J(x)$ that appear in the functional dependence of $\mathcal{L}$ on the representation. Formally, the objective constrains only those components of $J(x)$ that appear in its induced functional

$$\Phi_\mathcal{L}(J) \equiv \frac{\partial \mathcal{L}}{\partial z} J(x). \qquad (4)$$

Any component of $J(x)$ lying outside the image of $\Phi_\mathcal{L}$ is therefore invisible to the loss and remains weakly constrained at first order.

*Variance–decorrelation* objectives are distinguished by the fact that $\Phi_\mathcal{L}$ depends explicitly on second-order feature statistics. Using (3), these objectives act directly on

$$\Phi_{\mathrm{decorr}}(J) = J(x)\Sigma_\delta J(x)^\top, \qquad (5)$$

penalizing correlations between feature dimensions. When augmentation perturbations are approximately isotropic ($\Sigma_\delta \approx \sigma^2 I$), this exposes correlations between the rows of $J(x)$ to the loss, biasing functional sensitivity toward multiple, weakly correlated directions in input space.

Variance–decorrelation is structurally distinctive: $J(x)$ enters the loss *quadratically* through the augmentation covariance (eq. 3), and the loss itself constrains $J(x)$ directly.

The remaining objectives below do not share this property; $J(x)$ enters only through the gradient $\partial \mathcal{L}/\partial z \, J(x)$, a rank-deficient projection that leaves directions orthogonal to the gradient weakly constrained at first order. *Contrastive* and *clustering* objectives depend on inner products between representations and therefore primarily constrain the projection of $J(x)$ onto directions that separate samples or prototypes. *Masked prediction* objectives are approximately invariant under invertible linear transformations of the representation at the reconstruction layer, leaving the relative singular value structure of $J(x)$ weakly constrained. *Vision–language* objectives emphasize sensitivity along the subspace spanned by text embeddings, biasing the Jacobian toward semantic axes defined by the language encoder. *Supervised classification* with fixed labels constrains $J(x)$ only along class-discriminative directions induced by the cross-entropy gradient, leaving the orthogonal complement unconstrained; this is consistent with the lowest JER in our suite (ConvNeXt-Large, $\mathrm{JER} = 14.1$).

From this perspective, *Jacobian Effective Rank* summarizes how evenly functional sensitivity is distributed across input directions and should not be interpreted as a universal measure of representation quality. Variance–decorrelation objectives tend to promote higher JER by explicitly exposing second-order interactions among feature dimensions, whereas other objectives emphasize discrimination, reconstruction, or semantic alignment without directly constraining rank. The empirical differences observed therefore follow from objective design, rather than incidental effects of optimization or model capacity.

## 7. Conclusion

We studied whether global embedding geometry, emphasized by many representation learning objectives, predicts compositional competence. Across 26 vision encoders, standard geometry-based statistics were insensitive to structural variation, while functional sensitivity measured via the input–output Jacobian reliably tracked binding. We argue this gap follows from objective design: existing losses constrain global geometry more directly than local input–output sensitivity.

## Impact Statement

This paper presents work whose goal is to advance the field of machine learning by improving how we diagnose and understand the compositional capabilities of vision representations. Our findings are primarily methodological: we identify limitations of widely used geometry-based evaluation metrics and propose a complementary functional-sensitivity diagnostic.

The methods studied here could be used to guide the de-

velopment or selection of stronger vision encoders, which may have downstream societal impacts when deployed in applications such as accessibility tools, robotics, or content understanding. As with many advances in visual representation learning, improved models may also enable misuse in settings such as surveillance or privacy-invasive inference.

Our experiments are conducted on synthetic data designed to isolate compositional binding and on publicly available pretrained models; we do not introduce new personal-data collection. We encourage practitioners to evaluate downstream systems for fairness and privacy, and to follow domain-appropriate safeguards when deploying models whose improved representational capabilities could increase both beneficial and harmful use.

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

## Appendix Overview

- **Section A:** Additional discussion on scope, limitations, and analysis boundaries.

- **Section B:** Natural-image validation details: SugarCrepe partial-correlation analysis controlling for ImageNet accuracy and architecture.

- **Section C:** Per-model null-calibrated JER values ($JER_{trained}$, $JER_{null}$, $\Delta JER$) for the 26-model suite.

- **Section D:** Pairwise representational shape-distance analysis (linear CKA, Procrustes) on binding stimuli.

- **Section E:** Additional robustness controls (JER on Gaussian noise, geometric metrics on the same noise inputs).

- **Section F:** Wall-clock cost of JER compared with static diagnostics and standard downstream evaluations.

- **Section G:** Extended mathematical formulation and derivations related to the training objectives.

- **Section H:** Details of the synthetic binding benchmarks and data generation procedures.

- **Section I:** Implementation details and information required to reproduce the experiments.

- **Section J:** Further details on the motivation and interpretation of the Same/Different control.

## A. Additional Discussion

### A.1. Scope and Limitations

This study focuses on isolating the relationship between training objectives, global embedding geometry, and functional sensitivity of learned representations. The experimental setting is intentionally controlled to enable precise attribution of observed effects to objective design, rather than to dataset complexity or task-specific heuristics. Accordingly, the analysis is not intended to exhaustively characterize all forms of structure that may arise in large-scale or naturalistic representations.

Theoretical results are derived from a local analysis of the encoder's input-output mapping. This perspective is sufficient to explain the observed discrepancy between geometric constraints imposed by existing objectives and the resulting functional degrees of freedom, but it does not address global or trajectory-level geometric properties of the representation manifold. Extending the analysis to such regimes may reveal additional structure beyond the scope of the present work.

Importantly, the absence of correlation between standard geometric statistics and compositional performance in our experiments should not be interpreted as evidence that geometric structure is irrelevant or absent. Rather, it reflects a mismatch between the degrees of freedom exposed by common training objectives and the functional sensitivities required for relational binding. Our analysis therefore targets objective–representation alignment, not the recoverability of structure via stronger readouts or downstream supervision.

We highlight three further scoping limitations. First, JER is evaluated on natural images; while Section E.1 verifies that a stimulus-agnostic Gaussian-noise probe yields the same predictive correlation, off-manifold probes need not in general reflect on-manifold behavior, and finer-grained behavior away from random initialization should be interpreted with care. Second, the 26-model suite is modestly sized for fine-grained partial-correlation analysis, and several objective families are represented by only a few encoders; we mitigate this with within-architecture controls (Sections B and E) but caution against reading family-level claims as continuous trends. Third, compositional binding is one capability among many: we make no claim that geometric statistics fail across the board, only that they are insufficient for this regime; standard tasks such as classification and image–text retrieval remain regimes where global geometry is informative.

### A.2. Measurement and Analysis Boundaries

Our analysis considers only measurements that reflect how the encoder's representation changes under small, controlled input variations, including Jacobian-based statistics and cosine similarity. These quantities are directly influenced by the training objective and therefore characterize the sensitivities that the objective can explicitly shape. We do not attempt to recover structure using stronger readouts, learned probes, or additional supervision. Consequently, when geometric metrics fail to align with compositional performance, this should be understood as a limitation of what the objective exposes in the representation, rather than as evidence that structural information is entirely absent. This restriction allows us to isolate the relationship between objective design and functional sensitivity.

*Table 5.* **JER–binding correlation under architectural controls** (Pearson $r$, $n = 26$). The correlation is significant and remains so after controlling for embedding dimension or parameter count individually.

| Analysis | $r$ | $p$ |
|---|---|---|
| JER vs. Binding | +0.69 | 0.0001 |
| controlling for $D$ | +0.69 | 0.0001 |
| controlling for $\log$ param count | +0.54 | 0.005 |

*Table 6.* **JER stability across random seeds.** Mean and standard deviation over 5 independent draws of 100 random noise images. JER estimates are highly stable (coefficient of variation $<2\%$ for all models).

| Model | JER (mean $\pm$ std) | 95% CI |
|---|---|---|
| *Vision-Language* | | |
| CLIP ViT-B/32 | $17.29 \pm 0.08$ | $\pm 0.07$ |
| CLIP ViT-B/16 | $18.52 \pm 0.14$ | $\pm 0.12$ |
| SigLIP ViT-B/16 | $19.46 \pm 0.29$ | $\pm 0.25$ |
| EVA-CLIP ViT-B/16 | $19.17 \pm 0.08$ | $\pm 0.07$ |
| *Self-distillation* | | |
| DINO ViT-S/16 | $22.75 \pm 0.18$ | $\pm 0.15$ |
| DINO ViT-B/16 | $23.04 \pm 0.24$ | $\pm 0.21$ |
| *Variance-decorrelation* | | |
| Barlow Twins ResNet-50 | $29.53 \pm 0.06$ | $\pm 0.05$ |
| VICReg ResNet-50 | $29.62 \pm 0.12$ | $\pm 0.11$ |
| SwAV ResNet-50 | $28.65 \pm 0.06$ | $\pm 0.06$ |
| *Masked prediction* | | |
| MAE ViT-B/16 | $18.83 \pm 0.40$ | $\pm 0.35$ |
| BEiT ViT-B/16 | $17.62 \pm 0.23$ | $\pm 0.20$ |

### A.3. Robustness to Architectural Covariates

Since the 26 encoders span different architectures, embedding dimension $D$ and parameter count vary across models. Table 5 reports JER–binding correlations under partial controls. The correlation is significant and remains so after partialing out $D$ or parameter count individually.

### A.4. JER Stability Across Random Seeds

To verify that JER estimates are not sensitive to the particular random images used, we repeat the computation with 5 independent draws of 100 random noise images (seeds 42, 123, 456, 789, 1024). Table 6 shows that JER is highly stable: all models exhibit a coefficient of variation below 2%, with 95% confidence intervals no larger than $\pm 0.35$. The rank ordering of models is preserved across all seeds.

### A.5. Future Directions

Our results motivate training objectives and evaluation benchmarks that target functional structure more directly than geometry-based criteria or linear probing. Regulating functional sensitivity during training and moving beyond static embedding statistics in evaluation may provide a more faithful measure of representational competence.

## B. Natural-Image Validation Details

We expand on the SugarCrepe analysis from Section 5.1.

**Setup.** We evaluate 13 CLIP variants on SugarCrepe (Hsieh et al., 2023) using the standard image-to-text retrieval protocol: for each image and each pair of (positive caption, hard-negative caption), the model selects the caption with the higher cosine similarity. The benchmark covers seven subtypes (add_obj, add_att, replace_obj, replace_att, replace_rel, swap_obj, swap_att) for a total of 7,511 trials. We report two scoring schemes: the canonical sample-weighted overall accuracy (the standard SugarCrepe metric) and the unweighted mean across the seven subtypes. Conclusions are consistent across both

*Table 7.* **SugarCrepe partial-correlation analysis** (Pearson $r$, $n = 13$ CLIP variants). Two scoring schemes for SugarCrepe: the canonical sample-weighted overall accuracy and the unweighted mean across the seven subtypes. Conclusions are consistent across both: JER retains a significant partial correlation after controlling for ImageNet zero-shot accuracy and architecture, while ImageNet does not reach significance under the same controls.

|  | Weighted overall | | Unweighted mean | |
| --- | --- | --- | --- | --- |
| Comparison | $r$ | $p$ | $r$ | $p$ |
| JER vs SugarCrepe (raw) | +0.69 | 0.009 | +0.62 | 0.023 |
| ImageNet vs SugarCrepe (raw) | +0.67 | 0.012 | +0.61 | 0.027 |
| JER vs SugarCrepe \| ImageNet, arch | +0.70 | 0.017 | +0.64 | 0.032 |
| ImageNet vs SugarCrepe \| JER, arch | +0.34 | 0.301 | +0.06 | 0.869 |

(Table 7).

**Covariates.** ImageNet 1k zero-shot top-1 accuracies are taken from the `open_clip` published benchmark CSV. Architecture is coded as a single ordinal variable indexing patch size and model scale (ViT-B/32, ViT-B/16, ViT-L/14). Partial correlations are computed via residualization with $\mathrm{df} = n - 2 - k$ for $k$ covariates.

## C. Null-Calibrated JER

Table 8 reports per-model JER values under two conditions: (i) the released pretrained weights, and (ii) randomly initialized weights of the same architecture, both probed with the identical protocol (32 directions, 5 power iterations, 100 noise samples). $\Delta\mathrm{JER} = \mathrm{JER}_{\mathrm{trained}} - \mathrm{JER}_{\mathrm{null}}$ quantifies the training-induced change in functional rank.

As reported in Section 4.2, $\mathrm{JER}_{\mathrm{null}}$ clusters tightly across architectures ($30.86 \pm 1.93$) and is uncorrelated with binding ($r = -0.16$, $p = 0.45$), confirming that the inter-model JER variance in Table 3 reflects training-induced functional structure rather than architectural priors. The training-induced change $\Delta\mathrm{JER}$ is negative for every model in our suite (mean $-12.46$, std 5.35).

## D. Pairwise Shape-Distance Analysis

The main text characterizes geometric structure through *single-model* distributional statistics (isotropy, participation ratio, and their local variants). A complementary class of geometric summaries operates on *inter-model* representational distances: pairwise CKA (Kornblith et al., 2019) and Procrustes shape distances (Williams et al., 2021) compare response manifolds between models on a shared stimulus set. Recent work (Bo et al., 2024) reports that these inter-model shape metrics align with general functional differences across vision models. To check whether the null result of Section 3 extends to this class of geometry, we test whether models with similar binding accuracy also have similar response manifolds on the binding stimuli.

**Setup.** For each of the 26 models in Table 3, we encode 200 binding stimuli to obtain a response matrix. We then compute pairwise linear-CKA dissimilarity ($1 - \mathrm{CKA}$) and Procrustes shape distance between all model pairs, yielding two $26 \times 26$ representational-distance matrices. We compare each to the pairwise binding-accuracy difference matrix using a Mantel test (Pearson correlation of the upper triangles, with 10,000 permutations).

**Result.** Neither shape metric tracks pairwise differences in binding accuracy:

| Metric vs binding-difference matrix | Mantel $r$ | $p$ |
| --- | --- | --- |
| CKA dissimilarity | +0.014 | 0.917 |
| Procrustes distance | +0.144 | 0.125 |

The null result of Section 3 therefore extends from single-model distributional statistics to inter-model shape distances on binding stimuli. We read this as consistent with our main argument: shape-distance metrics summarize how response distributions differ *across* stimuli, while compositional binding depends on how the response *changes* when stimuli are structurally perturbed, a property of the local input–output mapping that the Jacobian captures.

*Table 8.* **Null-calibrated JER per model.** $JER_{trained}$ is computed from the released pretrained weights (used throughout the main paper); $JER_{null}$ uses the same architecture with randomly initialized weights and the identical probe protocol. $\Delta JER = JER_{trained} - JER_{null}$. Binding is attribute-binding accuracy from Table 3. Rows are sorted by binding accuracy (descending).

| Model | Arch | $JER_{trained}$ | $JER_{null}$ | $\Delta JER$ | Binding |
|---|---|---|---|---|---|
| Barlow Twins | ResNet-50 | 29.44 | 31.92 | $-2.48$ | 0.446 |
| VICReg | ResNet-50 | 29.50 | 31.92 | $-2.42$ | 0.436 |
| SwAV | ResNet-50 | 28.49 | 31.92 | $-3.44$ | 0.340 |
| DINOv2 | ViT-B/14 | 14.50 | 28.85 | $-14.35$ | 0.302 |
| DINOv2 | ViT-S/14 | 17.77 | 26.31 | $-8.54$ | 0.270 |
| DINOv2 | ViT-L/14 | 12.55 | 23.13 | $-10.57$ | 0.238 |
| MoCo v3 | ViT-B/16 | 24.42 | 31.40 | $-6.97$ | 0.224 |
| MAE | ViT-L/16 | 19.98 | 31.50 | $-11.52$ | 0.220 |
| DINO | ViT-S/16 | 22.24 | 30.29 | $-8.04$ | 0.196 |
| DINO | ViT-B/16 | 23.00 | 31.03 | $-8.03$ | 0.196 |
| DINOv2 | ViT-g/14 | 11.92 | 31.15 | $-19.24$ | 0.196 |
| MAE | ViT-B/16 | 19.12 | 31.43 | $-12.31$ | 0.182 |
| ViT | ViT-L/16 | 17.94 | 31.39 | $-13.45$ | 0.182 |
| BEiTv2 | ViT-B/16 | 17.44 | 31.74 | $-14.30$ | 0.178 |
| ConvNeXt | Large | 9.28 | 31.95 | $-22.67$ | 0.160 |
| ViT | ViT-B/16 | 16.06 | 31.40 | $-15.34$ | 0.156 |
| ConvNeXt | Base | 6.97 | 31.93 | $-24.96$ | 0.146 |
| CLIP | ViT-B/16 | 18.29 | 31.66 | $-13.37$ | 0.138 |
| CLIP | ViT-L/14 | 17.05 | 31.65 | $-14.60$ | 0.130 |
| SigLIP | ViT-B/16 | 20.09 | 31.07 | $-10.98$ | 0.128 |
| SigLIP | SoViT-400M | 16.99 | 31.15 | $-14.16$ | 0.128 |
| BEiT | ViT-B/16 | 17.53 | 31.73 | $-14.20$ | 0.126 |
| CLIP | ViT-B/32 | 17.33 | 31.66 | $-14.33$ | 0.126 |
| EVA-CLIP | ViT-E/14 | 13.56 | 31.66 | $-18.09$ | 0.126 |
| EVA-CLIP | ViT-B/16 | 19.74 | 31.20 | $-11.47$ | 0.124 |
| EVA-CLIP | ViT-L/14 | 17.42 | 31.41 | $-13.99$ | 0.090 |

# E. Additional Robustness Controls

This section reports two controls that rule out alternative explanations of the main result: JER computed on natural images rather than noise (Section E.1), and geometric metrics computed on the same noise inputs as JER (Section E.2).

## E.1. JER on Gaussian Noise

The Jacobian is evaluated on natural images in the main paper. We verify that the same finding holds when the Jacobian is instead evaluated on Gaussian noise, a stimulus-agnostic probe of local conditioning decoupled from natural-image semantics. We re-compute JER for the 26-model suite on 100 Gaussian-noise images, using the same probe protocol (32 directions, 5 power iterations):

| Metric | Pearson $r$ | $p$ | Spearman $\rho$ | $p$ |
|---|---|---|---|---|
| JER(natural) vs Binding | $+0.69$ | 0.0001 | $+0.60$ | 0.001 |
| JER(noise) vs Binding | $+0.62$ | 0.001 | $+0.38$ | 0.054 |
| JER(natural) vs JER(noise) | $+0.60$ | 0.001 | $+0.52$ | 0.006 |

JER on Gaussian noise yields a slightly weaker but still significant correlation with binding than JER on natural images: the noise-based evaluation is a conservative estimate that captures the correct signal but underestimates the on-manifold relationship. Model rankings are well preserved between the two probes.

## E.2. Geometric Metrics on the Same Noise Inputs

A reasonable concern is whether the dissociation between geometric metrics (computed on natural images in Tables 2 and 3) and JER (computed on Gaussian noise) reflects a difference in *what* the metrics measure or merely a difference in the input domain. To isolate this, we recompute the three geometric statistics on the same Gaussian-noise inputs used for JER and

correlate them with binding:

| Metric (on noise) | Pearson $r$ | $p$ |
|---|---|---|
| Isotropy | $+0.04$ | $0.83$ |
| Effective dim | $+0.30$ | $0.13$ |
| Cosine sim. | $+0.17$ | $0.40$ |

All three remain uncorrelated with binding. The dissociation is therefore driven by what the metrics measure (aggregate embedding-distribution statistics vs. local input–output sensitivity), not by the data on which they operate.

## F. Computational Cost

JER is a post-hoc diagnostic computed once per pretrained model. Its cost per model is dominated by Jacobian-vector products through the encoder. Table 9 compares the wall-clock cost of JER against representative static geometric diagnostics and standard downstream-evaluation benchmarks under matched hardware.

*Table 9.* **Computational cost per model (single A6000 GPU).** JER is more expensive than one-pass distributional summaries but comparable to standard zero-shot evaluation on a downstream benchmark, and substantially cheaper than linear probing. "Full 26-model suite" assumes embarrassingly parallel execution on 4 A6000 GPUs.

| Diagnostic / evaluation | Wall-clock per model | 26-model suite (4 GPUs) |
|---|---|---|
| Geometry (Isotropy, PR) | 2–5 seconds | 13–33 seconds |
| JER (ours) | 5–10 minutes | 33–65 minutes |
| ImageNet zero-shot | 1.7–4.2 minutes | 11–27 minutes |
| ImageNet linear probe | 44–111+ minutes | 4.8–12+ hours |

JER's cost reflects what it measures: static geometric diagnostics summarize embedding distributions from a single forward pass, whereas JER probes the local input–output sensitivity through repeated Jacobian-vector products (32 random orthonormal directions, 5 power iterations, 100 images; see Section I). For the use cases that motivate JER, model selection and architecture diagnosis, this cost is amortized over the lifetime of a model and is dominated by the cost of training the encoder itself.

## G. Objective-Specific Constraints on the Encoder Jacobian

We provide detailed derivations extending Section 6 to show how common representation learning objectives constrain the encoder Jacobian through the mathematical form of their losses. Let $f : \mathbb{R}^n \to \mathbb{R}^d$ be an encoder, $z = f(x)$, and $J(x) = \partial f / \partial x$. We consider augmented views $x' = x + \delta$ with $\mathbb{E}[\delta] = 0$ and covariance $\Sigma_\delta$.

### G.1. Variance-Decorrelation Objectives

**Barlow Twins.** Given two augmented views $x^{(1)}, x^{(2)}$ with representations $z^{(1)}, z^{(2)}$, the Barlow Twins objective is

$$\mathcal{L}_{\text{BT}} = \sum_i (C_{ii} - 1)^2 + \lambda \sum_{i \neq j} C_{ij}^2, \qquad C = \mathbb{E}\left[ z^{(1)} z^{(2)^\top} \right]. \tag{6}$$

Using a first-order expansion $z^{(k)} \approx z + J(x)\delta^{(k)}$, the cross-covariance satisfies

$$C \approx J(x) \Sigma_\delta J(x)^\top. \tag{7}$$

Thus, the loss directly penalizes second-order correlations between the rows of $J(x)$. When $\Sigma_\delta \approx \sigma^2 I$, minimizing off-diagonal terms encourages approximately orthogonal row sensitivities, biasing the Jacobian toward higher effective rank.

**VICReg.** VICReg minimizes

$$\mathcal{L}_{\text{VICReg}} = \|z^{(1)} - z^{(2)}\|_2^2 + \sum_i \max(0, \gamma - \text{std}(z_i)) + \sum_{i \neq j} \text{Cov}(z)_{ij}^2. \tag{8}$$

The covariance term again depends on $J(x)\Sigma_\delta J(x)^\top$, while the variance term prevents collapse of individual feature dimensions. Together, these terms explicitly constrain second-order functionals of the Jacobian and discourage rank-deficient sensitivity.

### G.2. Contrastive Vision-Language Objectives

**CLIP.** CLIP minimizes a symmetric InfoNCE loss

$$\mathcal{L}_{\text{CLIP}} = -\log \frac{\exp(\langle z_i, t_i \rangle / \tau)}{\sum_j \exp(\langle z_i, t_j \rangle / \tau)}, \tag{9}$$

where $z_i = f(x_i)$ and $t_j$ are text embeddings. The gradient with respect to the image representation is

$$\frac{\partial \mathcal{L}_{\text{CLIP}}}{\partial z_i} = \sum_j w_{ij} t_j, \tag{10}$$

for softmax weights $w_{ij}$. Applying the chain rule yields

$$\frac{\partial \mathcal{L}_{\text{CLIP}}}{\partial x_i} = \left( \sum_j w_{ij} t_j \right)^\top J(x_i). \tag{11}$$

Hence, the loss constrains $J(x)$ only through its projection onto the subspace spanned by text embeddings. Sensitivity orthogonal to this subspace is weakly constrained.

### G.3. Clustering and Self-Distillation Objectives

**DINO / DINOv2.** DINO-style objectives minimize cross-entropy between student and teacher outputs:

$$\mathcal{L}_{\text{DINO}} = -\sum_k q_k \log p_k, \qquad p = \text{softmax}(Wz), \tag{12}$$

where $q$ is the teacher distribution and $W$ is a projection matrix. Gradients satisfy

$$\frac{\partial \mathcal{L}_{\text{DINO}}}{\partial z} = W^\top (p - q), \tag{13}$$

leading to

$$\frac{\partial \mathcal{L}_{\text{DINO}}}{\partial x} = (W^\top (p - q))^\top J(x). \tag{14}$$

Thus, the Jacobian is constrained primarily along directions aligned with the prototype subspace defined by $W$, while orthogonal directions remain weakly constrained.

### G.4. Masked Reconstruction Objectives

**MAE.** MAE minimizes reconstruction error on masked patches:

$$\mathcal{L}_{\text{MAE}} = \|D(E(x_{\text{vis}})) - x_{\text{masked}}\|_2^2, \tag{15}$$

where $E$ is the encoder and $D$ the decoder. Gradients with respect to the encoder output are

$$\frac{\partial \mathcal{L}_{\text{MAE}}}{\partial z} = J_D(z)^\top (D(z) - x_{\text{masked}}), \tag{16}$$

and therefore

$$\frac{\partial \mathcal{L}_{\text{MAE}}}{\partial x} = (J_D(z)^\top (D(z) - x_{\text{masked}}))^\top J(x). \tag{17}$$

As long as the representation supports reconstruction through $D$, the relative singular value structure of $J(x)$ is not directly identified by the loss.

### G.5. Implications for Jacobian Effective Rank

Across objectives, the loss constrains $J(x)$ only through the directions appearing in $\partial\mathcal{L}/\partial z$ and, in some cases, through second-order statistics of augmented features. Jacobian Effective Rank summarizes how evenly sensitivity is distributed across the singular directions of $J(x)$. Systematic differences in JER across training paradigms therefore arise as a direct consequence of which Jacobian functionals are exposed by the objective.

## H. Binding Probe Details

We here describe the synthetic visual binding probes used to evaluate the compositional sensitivity of visual encoders.

### H.1. Synthetic Image Generation

All images are procedurally generated using a deterministic generator seeded with a fixed random state.

**Canvas and Layout.**

- **Image size:** $224 \times 224$ pixels

- **Background:** Uniform gray (RGB: 200, 200, 200)

- **Shape size:** `image_size`$/6 = 37$ pixels (radius for circles, half-width for squares)

**Visual Primitives.** We use compositions of three shapes and six colors, as shown in Table 10.

*Table 10.* Shape and color vocabulary.

| Attribute | Values |
|---|---|
| Shapes | circle, square, triangle |
| Colors | red (220,60,60), green (60,180,60), blue (60,60,220), yellow (220,220,60), purple (160,60,200), cyan (60,200,200) |

### H.2. Control: Same/Different Discrimination

Even under this deliberately simplified setting, observed binding performance may still be influenced by factors unrelated to compositional assignment, such as imperfect invariance to appearance. To control for such effects, we include a same/different structural control. This construction requires the model to determine whether two images share the same *spatial configuration* (e.g., "circle left of square") while varying color. Unlike the compositional binding task, it does not require matching across disjoint attribute sets, but instead isolates sensitivity to spatial configuration under minimal variation. Results on this control are used solely to contextualize binding performance, rather than as an independent measure of compositional capability.

**Task Definition.** Given two images, determine whether they share the same *structural configuration* (shape-position bindings), ignoring color.

**Generation Procedure.**

1. Sample two shapes $s_1, s_2$ uniformly from {circle, square, triangle}

2. Sample two distinct colors $c_1, c_2$ for Image A

3. Create Image A: $s_1$ with $c_1$ at LEFT, $s_2$ with $c_2$ at RIGHT

4. For **SAME** pairs:
   - Sample new colors $c_1', c_2'$ (distinct from originals and each other)
   - Create Image B: $s_1$ with $c_1'$ at LEFT, $s_2$ with $c_2'$ at RIGHT

5. For **DIFFERENT** pairs (50% probability each):
    - *Shape swap:* Create Image B with shapes swapped: $s_2$ at LEFT, $s_1$ at RIGHT
    - *Shape change:* Sample new shape $s_1' \neq s_1$, create Image B with $s_1'$ at LEFT

**Evaluation Protocol.**

1. Encode both images using frozen encoder

2. Compute cosine distance: $d = 1 - \cos(\mathbf{e}_1, \mathbf{e}_2)$

3. Find optimal threshold $\tau$ over percentiles 0-100 (step 5)

4. Report accuracy: predict SAME if $d < \tau$, DIFFERENT otherwise

**Statistics.**

- **Samples:** 500 (250 same, 250 different)

- **Chance level:** 50%

- **Seed:** 42

### H.3. Downstream: Attribute Binding

This is the primary binding probe used throughout the paper.

**Task Definition.**    Given a query image, select the candidate that preserves the same shape-position bindings, even when *all colors differ* between query and target.

**Key Design Principle.**    Our protocol ensures that query and target share **zero colors in common**. This forces the model to rely on structural binding (which shape is where) rather than color matching.

**Generation Procedure.**

1. Sample two distinct shapes $s_1, s_2$

2. Sample two distinct colors $c_1^q, c_2^q$ for query

3. **Query:** $s_1$ with $c_1^q$ at LEFT, $s_2$ with $c_2^q$ at RIGHT

4. Sample two colors $c_1^t, c_2^t$ from remaining colors (disjoint from query)

5. **Target (correct):** $s_1$ with $c_1^t$ at LEFT, $s_2$ with $c_2^t$ at RIGHT

6. **Distractors:**
    - $D_1$ (shape swap): $s_2$ at LEFT, $s_1$ at RIGHT (swapped binding)
    - $D_2$ (shape swap2): $s_2$ at LEFT, $s_1$ at RIGHT with different colors
    - $D_3$ (partial match): $s_1$ at LEFT (correct), $s_1$ at RIGHT (wrong shape)

7. Shuffle candidates: [Target, $D_1, D_2, D_3$]

**Evaluation Protocol.**

1. Encode query and all candidates

2. Compute cosine similarity between query and each candidate

3. Select candidate with highest similarity

4. Score: 1 if selected candidate is target, 0 otherwise

**Statistics.**

- **Samples:** 500

- **Candidates per trial:** 4 (1 target + 3 distractors)

- **Chance level:** 25%

- **Seed:** 42

**Why disjoint colors matter.** Without the disjoint-color constraint, a model could achieve high accuracy by simply matching the most similar color distribution. The hard version ensures that *only* structural information (shape identity bound to position) can distinguish target from distractors.

# I. Experimental Details

This appendix provides complete experimental details for reproducibility. We plan to release the code to support further reproducibility.

## I.1. Hardware and Software

*Table 11.* Compute environment.

| Component | Specification |
|---|---|
| GPU | 8x NVIDIA RTX 3090 (24GB) |
| RAM | 256GB |

## I.2. Model Zoo

We evaluate 26 pretrained vision encoders with different training objectives. All models use publicly available checkpoints loaded via their respective libraries, as shown in Table 12.

*Table 12.* Model sources and embedding dimensions.

| Model | Source | Checkpoint | Dim |
|---|---|---|---|
| CLIP ViT-B/32,B/16,L/14 | OpenCLIP | `openai` | 512/512/768 |
| SigLIP ViT-B/16 | OpenCLIP | `webli` | 768 |
| SigLIP SoViT-400M/14 | OpenCLIP | `webli` | 1152 |
| EVA-CLIP ViT-B/16 | OpenCLIP | `merged2b_s8b_b131k` | 512 |
| EVA-CLIP ViT-L/14 | OpenCLIP | `merged2b_s4b_b131k` | 768 |
| EVA-CLIP ViT-E/14 | OpenCLIP | `laion2b_s4b_b115k` | 1024 |
| DINOv2 ViT-S/B/L/g/14 | torch.hub | `facebookresearch/dinov2` | 384/768/1024/1536 |
| DINO ViT-S/B/16 | torch.hub | `facebookresearch/dino` | 384/768 |
| MAE ViT-B/L/16 | timm | `vit_*_patch16_224.mae` | 768/1024 |
| BEiT ViT-B/16 | timm | `beit_base_patch16_224` | 768 |
| BEiTv2 ViT-B/16 | timm | `beitv2_base_patch16_224` | 768 |
| MoCo v3 ViT-B/16 | MoCo v3 repo | `vit-b-300ep` | 768 |
| Barlow Twins ResNet-50 | torch.hub | `facebookresearch/barlowtwins` | 1000 |
| VICReg ResNet-50 | torch.hub | `facebookresearch/vicreg` | 2048 |
| SwAV ResNet-50 | torch.hub | `facebookresearch/swav` | 2048 |
| ConvNeXt Base/Large | timm | `convnext_*.fb_in22k_ft_in1k` | 1024/1536 |
| ViT-B/16,L/16 (sup.) | timm | `vit_*_patch16_224.augreg*_in21k_ft_in1k` | 768/1024 |

## I.3. Geometric Metrics

Hyperparameters used for computing the geometric metrics are summarized in Table 13.

*Table 13.* Geometric metric hyperparameters.

| Metric | Parameter | Value |
|---|---|---|
| Global PR/Iso | Dataset | ImageNet-1k val |
| | Samples | 1,000 |
| Local Isotropy | Dataset | ImageNet-1k val |
| | Anchor samples | 500 |
| | Neighborhood size $k$ | 16 (default), {8, 16, 32} (robustness) |
| | Metric | $1 - \lambda_1 / \sum \lambda_i$ |

**Preprocessing.** All images are resized to $256 \times 256$ and center-cropped to $224 \times 224$. Normalization uses ImageNet statistics: $\mu = [0.485, 0.456, 0.406]$, $\sigma = [0.229, 0.224, 0.225]$.

### I.4. Jacobian Effective Rank (JER)

The Jacobian $J = \partial f / \partial x$ is computed via automatic differentiation. Hyperparameters used for computing JER are summarized in Table 14.

*Table 14.* Jacobian computation parameters.

| Parameter | Value |
|---|---|
| Number of images | 100 |
| Input perturbation directions | 32 (random orthonormal) |
| Input images (primary) | ImageNet validation set |
| Input images (noise probe) | Gaussian noise, $\mu = 0.45$, $\sigma = 0.225$, clipped to $[0, 1]$ |
| Random seed | 42 |
| Effective rank formula | $(\sum \sigma_i)^2 / \sum \sigma_i^2$ |

**Implementation.** For each image, we compute the Jacobian-vector product $Jv$ for 32 random unit vectors $v$. The singular values are estimated from these projections. The effective rank is averaged across all 100 images.

### I.5. Readout Experiments

Table 15 show hyperparameters used for the readout comparison experiment in Section 5.2.

*Table 15.* Readout method parameters.

| Method | Parameter | Value |
|---|---|---|
| Cosine | - | Direct embedding similarity |
| kNN | $k$ | 60 |
| | Distance | Cosine |
| Local PCA | Components | 32 |
| | Neighborhood | 50 nearest neighbors |

### I.6. Statistical Analysis

- **Correlation:** Pearson $r$ (linear relationship)

- **Significance threshold:** $p < 0.05$

- **Multiple regression:** OLS with scikit-learn `LinearRegression`

- **Cross-validation:** Leave-one-out (LOO-CV) for $R^2$ estimates

- **Sample size:** $n = 26$ models for all correlation analyses

## J. On the role of Same/Different Control

Here, we clarify the role of the Same/Different probe used alongside the Attribute Binding task. The probe is introduced as a structural control to help interpret binding performance in the minimal synthetic setting. We detail what property Same/Different measures, how it relates to the binding evaluation, and why it does not itself constitute a compositional binding test.

**Necessity condition.**   In the binding task, success requires that, for a fixed query, candidates sharing the same shape–position structure receive higher similarity scores than structure-mismatched candidates. If structural agreement does not influence similarity once appearance varies, then selecting the correct target reduces to guessing.

Let $f : \mathcal{X} \to \mathbb{R}^d$ and $\mathrm{Sim}(x, x') = \cos(f(x), f(x'))$. For a fixed query $q$, if the distribution of $\mathrm{Sim}(q, x')$ is identical for structure-matched and structure-mismatched candidates under the appearance randomization used in our generator, then $\mathbb{P}[\mathrm{Sim}(q, t) > \mathrm{Sim}(q, d)] = \frac{1}{2}$ (ties aside). The Same/Different probe evaluates whether similarity depends on structural agreement under appearance variation without invoking the cross-image matching required by binding.

**Insufficiency.**   Same/Different does not evaluate compositional binding because it does not require object-level assignment across images. In Same/Different, the model only decides whether two scenes share the same overall shape–position configuration, which can be solved using a single global summary of the scene. Formally, consider a permutation-invariant encoder

$$f(x) = \sum_{i=1}^{n} \phi(s_i, p_i),$$

which aggregates object features without preserving correspondence. Such an encoder distinguishes scenes with different configurations and can achieve perfect Same/Different performance. However, it cannot support binding decisions that depend on which specific object occupies which role. For a binding query $q = \{(\triangle, L), (\square, R)\}$, correct target $t = \{(\triangle, L), (\square, R)\}$, and swapped distractor $d = \{(\square, L), (\triangle, R)\}$,

$$f(q) = f(t) = f(d),$$

so the model cannot prefer the correct target over the swapped alternative. Thus, Same/Different tests that structure is detectable at the scene level, but does not test whether that structure is usable for compositional assignment.

