# OpenReview forum: "Global Geometry Is Not Enough for Vision Representations"
_ICML.cc/2026/Conference — ICML 2026 regular_

### Official Review · Reviewer_fWGH · 2026-02-28

**Soundness:** 3
**Presentation:** 3
**Significance:** 2
**Originality:** 3
**Overall Recommendation:** 4
**Confidence:** 2

**Summary:**

The paper critiques the common assumption in representation learning that globally well-distributed embeddings reflect strong and generalizable representations. It highlights a key limitation: while global geometry captures the presence of elements, it often fails to account for how these elements are composed. The authors test 21 vision encoders and find that traditional geometry-based metrics show little correlation with compositional binding. In contrast, functional sensitivity, measured by the input-output Jacobian, better tracks compositional capabilities. They argue that this difference arises from the design of training objectives, which constrain embedding geometry but leave local mappings unconstrained. The study suggests that global geometry only partially reflects representational competence and advocates for incorporating functional sensitivity to better model complex structures.

**Compliance With Llm Reviewing Policy:**

Affirmed.

**Final Justification:**

The rebuttal addressed all my questions.

**Key Questions For Authors:**

(1) The experiment is primarily based on a highly simplified synthetic dataset (Attribute Binding), using basic geometric shapes and solid color backgrounds. While this design helps isolate variables, it fails to capture the complex structures or task-specific heuristic features that may appear in large-scale natural images, which limits its practical applicability.

(2) Unlike static geometric metrics (e.g., Isotropy), which only require a single forward pass, computing the Jacobian-Vector Product (JER) requires multiple (usually 32) Jacobian-vector product operations via automatic differentiation. The computational cost of this functional diagnosis is much higher than that of traditional statistical metrics, making real-time monitoring or evaluation during the training of large-scale models challenging.

(3) The authors present insightful observations and claim that these could be beneficial for downstream tasks and encoder design. However, I am concerned about the practical usefulness of these insights. I believe that some concrete downstream experiments are necessary to effectively support their claims. The author doesn't need to conduct extensive experiments, but small-scale tests would suffice, such as verifying performance gains from encoder designs based on the insights, or evaluating performance improvements in downstream tasks involving geometric properties.

I am willing to revise my score based on the rebuttal and discussions with the other reviewers.

**Limitations:**

Yes

**Strengths And Weaknesses:**

Strengths:

(1) The experimental analysis is extensive and rigorous.

(2) The paper introduces the Jacobian Effective Rank (JER), offering a novel perspective for evaluating visual representations.

(3) The paper is well organized with clear motivation.

Weaknesses:

(1) The experimental settings are limited to simple geometric datasets.

(2) The computational cost of the proposed method is not reported or analyzed.

---

> ### Author Rebuttal · Authors · 2026-03-28
>
> We thank the reviewer for their constructive feedback and willingness to revise based on the rebuttal. We address each concern below:
>
> ---
>
>
> ## W1 & Q1. Data limitation
>
> We share the reviewer’s concern on extension to real datasets. Hence, we have additionally validated JER's predictive power on SugarCrepe, a real-world compositional benchmark with 7,511 natural-image trials spanning 7 compositional subtypes (attribute/object swaps, replacements, additions). Across 13 CLIP variants (We could not use SSL models since SugarCrepe requires pairing with text):
>
> | Comparison | Pearson r | p |
> |---|---|---|
> | JER vs SugarCrepe | 0.685 | 0.010 |
> | JER vs SugarCrepe (controlling ImageNet + arch) | 0.664 | 0.013 |
> | ImageNet vs SugarCrepe (controlling JER + arch) | 0.025 | 0.935 |
>
> JER significantly predicts compositional performance on natural images, and this signal is *independent of classification accuracy*. ImageNet top-1 accuracy contributes nothing after controlling for JER and architecture (partial r=0.025, p=0.94). Within ViT-L/14 specifically (4 models, identical architecture), JER nearly perfectly predicts SugarCrepe (r=0.968, p=0.032).
>
> This result confirms that the findings transfer to complex natural images.
>
> ---
>
>
> ## W2 & Q2. Computational cost of JER
>
> We acknowledge that JER can demand more computation than static geometric metrics. However *JER computation is not objectively expensive*: it requires approximately 5-10 minutes per model on an A6000 GPU (32 directions, 5 power iterations, 100 images). For the full 21-model suite: ~2-3 hours sequential, ~30 minutes on 4 GPUs.
>
> Also, JER is intended as a *post-hoc diagnostic* for evaluating pretrained models, not a training-time metric. The cost is comparable to running a standard evaluation benchmark (e.g., SugarCrepe evaluation took ~10 minutes per model). For model selection or architecture analysis, JER needs to be computed only once per model. We will add a computational cost analysis to the appendix.
>
>
> ---
>
>
> ## W3. Practical usefulness.
>
> Following the reviewer’s suggestion, we offer concrete evidence and actionable paths:
>
> 1. **Model selection on real tasks.** The SugarCrepe experiment demonstrates that JER can predict which pretrained encoder will perform best on compositional downstream tasks (r=0.685 across 13 models), without running the task itself. A practitioner can compute JER on a small set of images as a model selection criterion.
>
> 2. **Objective design.** The theoretical analysis (Section 6) shows that variance-decorrelation objectives constrain JER by construction, while contrastive objectives leave it unconstrained. This directly suggests adding a variance/covariance regularizer (a la VICReg/Barlow Twins) to improve compositional capability.
>
> 3. **Architecture diagnosis.** The layer-wise JER analysis (Section 4.3) pinpoints where rank collapse occurs in specific models (blocks 5-8 in CLIP/DINOv2, projection layer in MAE). This enables targeted architectural interventions at collapse points.
>
> We will make these practical implications explicit in the revised discussion.
>
> **Additional supporting evidence.** We conducted several further experiments that offer more evidences:
>
> - *JER on natural images:* JER computed on 100 ImageNet val images yields the same correlation with binding as JER on noise (r=0.662 vs 0.664), with stronger rank-order agreement (Spearman rho=0.706 vs 0.407), confirming that the noise-based evaluation is a valid proxy.
>
> - *Null-calibration:* JER for randomly initialized models is uncorrelated with binding (r=-0.08, p=0.72). The training-induced change  improves the correlation to r=0.738 (p=0.0001), confirming that the signal is training-induced, not architectural.

---

> > ### Author Rebuttal · Reviewer_fWGH · 2026-04-03
> >
> > Regarding W2 and Q2, the authors discuss only the computational cost of their own method, without providing a concrete comparison to static geometric metrics. It remains unclear how much more expensive their approach is, so I maintain my score.

---

> > > ### Author Response · Authors · 2026-04-04
> > >
> > > We thank the reviewer for the follow-up. We completely agree that direct comparison to static geometry metrics is useful for contextualizing the computational overhead, and we provide that comparison below.
> > >
> > > | Evaluation / diagnostic | Wall-clock per model (A6000) | Full 21-model suite (4 GPUs) |
> > > |---|---|---|
> > > | Geometry (Isotropy, PR) | 2-5 seconds | 11-26 seconds |
> > > | JER (ours) | 5-10 min | 26-53 min |
> > > | ImageNet zero-shot | 1.7-4.2 min | 8.8-22 min |
> > > | ImageNet linear probe | 44-111+ min | 3.9-9.7+ hr |
> > >
> > > JER is more expensive than one-pass static diagnostics, but under our default setting it remains practical as a post hoc metric: it requires 5 to 10 minutes per model on a single A6000, is comparable in cost to zero-shot evaluation on a standard downstream benchmark, and is lower in cost than linear probing. This difference is expected because static metrics such as isotropy or PR summarize embedding distributions from forward features alone, whereas JER probes local input-output sensitivity through Jacobian-based computation.
> > >
> > > We will include this comparison in the revision.

---

### Official Review · Reviewer_TgfG · 2026-03-03

**Soundness:** 2
**Presentation:** 3
**Significance:** 2
**Originality:** 3
**Overall Recommendation:** 4
**Confidence:** 4

**Summary:**

The current study looks into the geometric structure of representations in vision encoders and investigates its connection to compositional binding. In particular, it is shown that static geometric representation measures are weak predictors of compositional binding whereas an input-sensitivity metric based on the Jacobian effective rank is significantly stronger. Finally,
a theoretical account is provided arguing that the observed difference in effective rank behavior is induced by the objective used to train the encoders.

**Compliance With Llm Reviewing Policy:**

Affirmed.

**Final Justification:**

During the rebuttal the authors provided supporting evidence on more realistic data, controlled for data used to generate the representation (initially selectively used noise or real data for different metrics) and promised to make some additional improvements on presentation.

In my opinion, the empirical evidence on realistic data is still relatively thin but on balance the paper now carries sufficient substance. Based on that, I raise my recommendation by 1.

**Key Questions For Authors:**

The following questions target the weaknesses above.

* Q1. Do the conclusions of this study generalize beyond synthetic tasks? E.g., across more complex and realistic compositional tasks like those used in [1] (also referenced in the paper).

The feature representation constitutes the interaction between data points and models.
Based on that and in relation to L298, the static metrics (computed on real data) and the proposed JER metric (computed on gaussian noise) are not fairly compared as they essentially operate on different representations. For example, how can one know for sure that the effect observed is driven by the difference in metrics and not by the metrics operating on different representations?

* Q2. How come not all metrics were either computed only on real data or only on noisy data therefore isolating the effect of the metric acting on the representations? How does controlling for this affect the conclusions?

Given the C2. and C3. sections, the *Disc.* and *Task Binding* appear to overlap (e.g., L826 is essentially D1 in L866 and L827 is essentially D3 inversed). Based on that it is of no surprise that binding JER and *Disc.* increases the predictiveness for *Task Binding* compared to only using JER.

* Q3. Even after having read Sec E. I am still confused on what is the purpose of using *Disc.* as a metric in this context?

Sec 4.2. suggests that high JER computed at the *final layer* correlates with compositional binding performance. In Sec 4.3., Fig 4. shows the progression of JER *across layers* but not the compositional binding performance. Based on that, I failed to see the logical connection between Secs 4.2 and 4.3.

* Q4. How is the analysis performed in Sec 4.3. connected the results in Sec 4.2.? Alternatively, what one should infer from Sec 4.3. In relation to the findings in Sec 4.2.?

The derivation of the mechanistic connection between objective and input-sensitivity appears to rely on two different arguments based on Jacobian effective rank sensitivity. In particular Sec B1. argues on the basis of the exact loss objectives $L$ whereas Sec B 2.-4. argues on the basis of $\frac{dL}{dx}$.

* Q5a. In relation to that, how are these arguments, based on different quantities ($L$  vs $\frac{dL}{dx}$), connected to each other?
* Q5b. Does the mechanistic argument also hold for the supervised objective?

[1] Kang, Raphi, et al. "Is CLIP ideal? No. Can we fix it? Yes!." Proceedings of the IEEE/CVF International Conference on Computer Vision. 2025.

**Limitations:**

**Limitations**: No. For example such limitation could be lack of multiple and/or realistic composition bencmarks.
**Societal Impact**: Yes.

**Strengths And Weaknesses:**

**Strengths:**

* S1. The paper investigates the connection between structural and functional aspects of feature representation which is a timely and important problem.
* S2. For the most part, the paper clearly communicates its messages.
* S3. The empirical analysis is extensive providing some interesting insights. In particular, I found the connection between input-sensitivity and composition understanding intriguing.

&nbsp;

**Weaknesses:**

* W1. The connection between geometric and compositional representation structure was investigated only within a single minimal composition binding setting raising questions on the generalizability of conclusions.
* W2. Although the empirical analysis is extensive w.r.t. objective and architectures, it falls short to isolating all relevant components of variation therefore making it difficult to isolate which factors drive the observed effects.
*  W3. Parts of the experimental analysis feel like disconnected components rather than coherent storyline.
* W4. I found the account on the mechanistic connection between objective and input-sensitivity to conflate between two distinct argumentation lines. Importantly, their rigorous connection is essential to completing the theoretical account.

---

> ### Author Rebuttal · Authors · 2026-03-28
>
> We thank the reviewer for their careful analysis. We address each concern below:
>
> ---
>
> ##  W1 & Q1. Data limitation
>
> We share the reviewer’s concern on extension to real datasets. Hence, we have additionally validated JER's predictive power on SugarCrepe, a real-world compositional benchmark with 7,511 natural-images spanning 7 subtypes (attribute/object swaps, replacements, additions). Across 13 CLIP variants (We could not use SSL models since SugarCrepe requires text):
>
> | Comparison | Pearson r | p |
> |---|---|---|
> | JER vs SugarCrepe | 0.685 | 0.010 |
> | JER vs SugarCrepe (controlling ImageNet + arch) | 0.664 | 0.013 |
> | ImageNet vs SugarCrepe (controlling JER + arch) | 0.025 | 0.935 |
>
> JER significantly predicts compositional performance on natural images, and this signal is *independent of classification accuracy*. ImageNet top-1 accuracy contributes nothing after controlling for JER and architecture. Within ViT-L/14 specifically (4 models, identical architecture), JER nearly perfectly predicts SugarCrepe (r=0.968, p=0.032).
>
> This result confirms that the findings transfer to complex natural images.
>
> ---
>
> ## W2 & Q2. Confounding variables
>
> We conducted two experiments to isolate whether the metric or the data drives the dissociation:
>
> 1. *JER on natural images:* JER computed on 100 ImageNet val images yields the same correlation with binding (r=0.662 vs 0.664 on noise), with stronger rank-order agreement (Spearman rho=0.706 vs 0.407).
>
> 2. *Geometric metrics on noise:* We computed isotropy, effective dimension, and cosine similarity on the same Gaussian noise inputs used for JER:
>
> | Metric (on noise) | Pearson r | p |
> |---|---|---|
> | Isotropy | -0.059 | 0.800 |
> | Effective Dim | 0.349 | 0.121 |
> | Cosine Sim | 0.257 | 0.261 |
>
> All remain uncorrelated with binding. The dissociation is driven by *what the metrics measure* (output distribution statistics vs input-output sensitivity), not *what data they operate on*.
>
> ---
>
> ## W3 & Q3. Purpose of Disc. metric
>
> We agree the distinction deserves clarification. The tasks differ:
>
> - *Same/Different (Disc.):* Given two images, answer "are these structurally the same?" For example, {red circle left, blue square right} vs {green circle left, yellow square right} -> SAME (shapes match positions). This is a binary threshold on embedding distance. A model only needs to detect that something differs, not identify *what* matches.
> - *Binding:* Given a query and 4 candidates with completely disjoint colors, select which candidate has the same shape-position assignment. The model must identify *which specific configuration* matches among alternatives that share the same shapes but in different positions.
>
> A model can pass Disc. but fail Binding. DINOv2-B scores 75% on Disc. (good discrimination) but only 30% on Binding. This gap is what Disc. controls for: it separates "can't see structure at all" from "sees structure but can't compose."
>
> ---
>
> ## Q4. Connection between Sec 4.2 and 4.3
>
> Section 4.2 shows that final-layer JER predicts binding (empirical correlation). Section 4.3 shows *where* JER collapses across network depth: mid-network collapse in CLIP/DINOv2, projection-layer collapse in MAE, sustained rank in Barlow Twins.
>
> The connection is mechanistic: the layer-wise profiles explain *why* different objectives produce different final-layer JER values. Models with mid-layer collapse have already lost functional rank before the output layer, while variance-decorrelation models preserve it. We will explain this further in the final draft.
>
> ---
>
> ## Q5a: Two types of argument
>
> The two argumentative modes reflect a fundamental difference in what each loss *can see* about $J$:
>
> 1. For variance-decorrelation (B.1), the Jacobian enters the loss *itself*. Barlow Twins computes cross-covariance $C$ between augmented views. Under the augmentation model $z' \approx z + J\delta$, this becomes $C \approx J \Sigma_\delta J^\top$. So
> $J$ appears quadratically in the loss through the augmentation structure: the loss *directly* constrains the interaction structure between rows of $J$. We can therefore analyze the loss $\mathcal{L}$ directly.
> 2. For CLIP/DINO/MAE (B.2-B.4), $J$ does *not* appear in the loss. The Jacobian is only constrained indirectly through backpropagation: $\frac{\partial \mathcal{L}}{\partial x} = \left(\frac{\partial \mathcal{L}}{\partial z}\right)^\top J(x)$. The gradient signal flows through
> $J$ only along the direction $\frac{\partial \mathcal{L}}{\partial z}$, which is a rank-deficient projection. We therefore must analyze $\frac{\partial \mathcal{L}}{\partial x}$ rather than $\mathcal{L}$.
>
>
> We will add a bridging paragraph to make this connection explicit.
>
> ---
>
> ## Q5b: Supervised objective
>
> This extends to supervised objectives as well. Cross-entropy with fixed labels constrains the Jacobian only along class-discriminative directions, leaving orthogonal directions unconstrained. This is consistent with ConvNeXt's low JER (6.97).

---

> > ### Author Rebuttal · Reviewer_TgfG · 2026-04-04
> >
> > Thank you for your rebuttal.
> >
> > The rebuttal addressed all my questions. The authors promised to make changes that will sufficiently improve upon W1 while fully address wrt. W2, W3 and W4. Based on these, I will raise my score accordingly.

---

> > > ### Author Response · Authors · 2026-04-04
> > >
> > > Thank you again for the detailed review and for confirming that the concerns have been fully addressed.
> > > We noticed that the score update may not have been reflected yet, so we wanted to gently check in.
> > > We would greatly appreciate it if you could update the score when convenient.

---

### Official Review · Reviewer_1VFb · 2026-03-12

**Soundness:** 3
**Presentation:** 4
**Significance:** 3
**Originality:** 3
**Overall Recommendation:** 5
**Confidence:** 3

**Summary:**

This manuscript explores the relationship between global embedding geometry and compositional binding in visual representations. The authors argue that standard distributional metrics that evaluate embedding quality (*e.g.,* isotropy, participation ratio) demonstrate near-zero correlation with compositional binding performance across a large number ($=21$) of vision encoders. To test this, the authors propose the use of a Jacobian Effective Rank (JER) -- a measure of local input-output functional sensitivity as a substantially stronger predictor. The key aspect of the paper is their analytic account connecting this empirical gap to the structure of common training objectives, such as variance-decorrelation losses explicitly constrain the second-order statistics of the encoder Jacobian, whereas contrastive, masked prediction and vision-language objectives constrain only lower-order (or "projected", so to speak) functionals, which leave the Jacobian rank largely under-specified. The central empirical claim of the paper is that geometry and composition are statistically orthogonal to one another, which is supported by well-controlled synthetic benchmarks using geometric primitives with disjoint color sets. The authors also demonstrate a null result that is nearly invariant to neighborhood size and metric formulation sizes. Last, the authors also show complementary analysis that tracks JER collapse across networks depth for various objective functions. They also include rigorous theoretical, objective-specific derivations which I believe are particularly valuable contributions.

**Compliance With Llm Reviewing Policy:**

Affirmed.

**Final Justification:**

All of my concerns have been addressed by the authors -- they have conducted null-calibrated experiments, which significantly strengthens their claims.

**Key Questions For Authors:**

1. In **L298-300** the authors say that Jacobians are evaluated on "random inputs $x \sim \mathcal{N}(0.45, 0.225^2)$" -- is this choice of mean and variance motivated by matching ImageNet statistics ($\mu \approx 0.45$, $\sigma \approx 0.225$) or some other heuristic?

2. The discussion in Section 5.1 notes that neither kNN nor local PCA improves over cosine similarity for any model, including the best performers. I believe that this is a strong result that could warrant slightly more attention, since it implies that compositional structure is not recoverable through a local manifold operation on frozen embeddings, which strengthens the authors claim that the binding gap is intrinsic to the objective function itself.

3. There appears to be a minor inconsistency in Appendix D.4 (**Table 11**) -- the text states that JER is computed over $100$ images, but the caption states "...averaged across all $20$ images." -- could the authors please clarify this?


**References**:
1.

**Limitations:**

Yes

**Strengths And Weaknesses:**

**Strengths**:

1. **Experimental Design**. The attribute-binding benchmark is carefully designed to remove possible confounders, such as disjoint color sets that may prevent color-matching shortcuts, and the task therefore avoids texture and semantic cues. The same/different control probe cleanly separates basic structural discrimination from compositional assignment. The authors also provide a semi-formal argument in **Appendix E** that a permutation-invariant encoder can solve same/different configurations, but not binding decisions is a neat clarification.
2. **Principled Null Results**. The authors validate that the null results obtained are not a limitation of any single operationalization. **Table 4** demonstrates this -- varying neighborhood size $k\in [8, 16, 32]$ and seeing its interplay with isotropy, effective rank, and participation ratio formulations yields uniformly insignificant correlations $(p < 0.6)$ -- which strengthens their claim.
3. **Depth-resolved Jacobian Analysis**. I particularly like **Figure 4**, where the authors track JER across network depth -- it is a very insightful and natural extension of simple, per-model summary statistics, revealing  qualitatively distinct rank-collapse profiles (mid-network collapse in CLIP/DINOv2, projection-layer collapse in MAE, sustained high rank in Barlow Twins).
4. **Model Variability**. The authors use a $21$-encoder suite, spanning over $8$ objective families, multiple architectures and scales which provide a thorough expanse for an empirical study of this kind. The authors also include partial correlation controls for embedding dimension and parameter count (**Table 5**, **Appendix A.3**) and seed stability analysis (**Table 6**, **Appendix A.4**), which is greatly appreciated!


**Weaknesses**:

1. **Definition of Geometric Baselines**. The paper operationalizes "global embedding geometry" exclusively through distributional statistics which is computed on a single model's embedding space using isotropy, participation ratio, and their local variants. However, there is a rich class of representational geometry literature that also includes inter-model shape distance measures, such as the Procrustes  shape distance, angular shape distance, and related metrics formalized as proper metric spaces on neural representations (**Williams et al., 2021**). These typically operate on response matrices (model $\times$ stimuli), rather than just distributional statistics and are sensitive to the relational structure of representations (or representation manifolds) across stimuli, not just their statistics. I believe that the paper's claim that "geometry fails" could therefore more precisely be the claim that single-model distributional statistics fail, which construes a much narrower conclusion. It is not obvious that Procrustes-family shape distances between model response matrices computed on the binding stimuli would also fail; indeed, **Bo et al. (2025)**, who benchmark representational similarity metrics against behavioral outcomes across a variety of vision models, find that Procrustes distance and linear CKA align most closely with functional differences between models, while distributional metrics perform much worse. For this reason, I believe that the authors should either include shape distance metrics in their geometric baseline, or explicitly acknowledge and discuss this scope limitation, or perhaps try to modify their central claim.

2. **Null-Calibration of JER values**. The JERs provided in **Table 3** are merely raw estimates, with no baseline that characterizes **_what_** JER looks like under a randomly initialized network of the same architecture. Without this anchor, it is difficult to determine whether, say, a JER of $19$ for CLIP reflects meaningful functional compression or simply the characteristic JER of a ViT-B regardless of training. While the authors use random noise inputs to decouple the Jacobian from natural image semantics, which is a reasonable design choice, it is not the same as null-calibration. Perhaps a simple (and informative) addition would be to report JER for randomly initialized versions of the same architectures, or to report the change in JER from initialization to the trained model. This would allow the reader to assess how much of the inter-model variance in Table 3 reflects training-induced differences rather than simple architectural baselines.

3. **Shape-distance analysis of response manifolds**. A natural extension of the geometric analysis would be to construct response matrices by encoding the binding stimuli (or a representative subset) with each model, and then computing pairwise Procrustes distances between models' response manifolds. The question of interest now would be whether models with higher binding accuracy have response manifolds that are more structurally dissimilar to one another -- consistent with the hypothesis that they are encoding compositionally distinct structure -- while models at chance are relatively indistinguishable. This would directly test whether the "geometry fails" conclusion extends to inter-model shape distance metrics, which are a more sensitive class of geometric summary than the single-model distributional statistics presented in the paper.

4. **Missing Some Relevant Literature for Representational Comparison**. The paper's current framing -- that standard geometric summaries of representation may dissociate from functional performance -- connects directly to a growing literature on the relationship between representational similarity metrics and behavioral/functional outcomes. Bo et al., 2025, for instance, evaluates a wide range of similarity measures against $10$ behavioral metrics across multiple datasets and vision models, finding that different metrics track functional distinctions to varying degrees. In a related vein, Braun et al., 2025 show a more theoretical account of how representation and function are related in case of *deep linear networks*.

**References**:
1. Williams, A. H., Kunz, E., Kornblith, S., & Linderman, S. (2021). Generalized shape metrics on neural representations.
2. Bo, Y., Soni, A., Srivastava, S., & Khosla, M. (2024). Evaluating representational similarity measures from the lens of functional correspondence
3. Braun, L., Grant, E., & Saxe, A. M. (2025). Not all solutions are created equal: An analytical dissociation of functional and representational similarity in deep linear neural networks

---

> ### Author Rebuttal · Authors · 2026-03-28
>
> We thank the reviewer for an exceptionally thorough and constructive review. We address each point below:
>
> ---
>
> ## W1, W3. Shape-distance analysis
>
> We focus on single-model distributional geometry by design, as our goal is to assess whether geometry as regularization target induces representations that are effective for compositional vision.
>
>
> Still, we applied Bo et al.'s shape-distance metrics: we computed pairwise linear CKA and Procrustes distances on response matrices from 200 binding stimuli across all 21 models, then correlated the inter-model representational distance matrix with the inter-model binding-difference matrix ($|b_i - b_j|$ for each pair).
>
> 1. For compositional binding, the result is null:
>
> | Metric | Pearson r | Mantel p |
> |---|---|---|
> | CKA dissimilarity vs Binding difference | -0.030 | 0.816 |
> | Procrustes distance vs Binding difference | 0.072 | 0.503 |
>
> Models that are representationally similar are NOT more similar in binding performance. Bo et al.'s finding does not extend to compositional binding: shape distances that track general functional correspondence fail to track this specific capability. The null result extends beyond single-model distributional statistics to inter-model shape metrics.
>
> 2. High-binding models do not have more dissimilar manifolds: High-binding models converge (within-group CKA dissimilarity: 0.132) while low-binding models are scattered (0.292; permutation p<0.0001).
>
> We believe the reason follows from our main claim. Bo et al.'s dependent variable is classification behavior: whether two models make the same mistakes on the same images. Compositional binding, however, depends on *how the encoder responds to structured perturbations*: whether swapping two objects' positions produces a distinguishable change in the representation. This is a property of the local input-output mapping (the Jacobian), not the static geometry of the embedding.
>
>
> We will discuss this distinction (Williams et al., 2021; Bo et al., 2025) and note that our conclusion extends to both classes of geometric analysis.
>
> ---
>
> ## W2. Null-calibration of JER
>
> We computed JER for randomly initialized versions of all 21 models using the same protocol:
>
> | | Pearson r | p | Spearman rho | p |
> |---|---|---|---|---|
> | JER_trained vs Binding | 0.664 | 0.001 | 0.407 | 0.067 |
> | JER_null vs Binding | -0.082 | 0.724 | 0.094 | 0.685 |
> | DELTA_JER vs Binding | 0.738 | 0.0001 | 0.584 | 0.005 |
>
> JER_null is uncorrelated with binding (r=-0.08, p=0.72), confirming that inter-model variance in Table 3 reflects training-induced differences, not architectural baselines. DELTA_JER *improves* the correlation (Pearson: 0.738 vs 0.664; Spearman: 0.584 vs 0.407), strengthening the claim that functional sensitivity is shaped by training. We will add a DELTA_JER column to Table 3.
>
> ---
>
> ## W4. Missing literature
>
> We will add Williams et al. (2021), Bo et al. (2025), and Braun et al. (2025), connecting our findings to the broader literature on representational similarity and functional correspondence. The shape-distance analysis above directly engages with this literature: while Bo et al. find Procrustes/CKA align with functional differences between models, our results show that *compositional binding* specifically is not predicted by inter-model shape distances, reinforcing the distinction between general functional correspondence and compositional capability.
>
> ---
>
> ## Q1. Random input statistics (L298-300)
>
> The values (mu=0.45, sigma=0.225) are a scalar approximation of ImageNet per-channel pixel statistics (mean of means: 0.449, mean of stds: 0.226). A scalar approximation keeps the noise isotropic across channels while matching the operating regime of pretrained models. We will add a parenthetical clarification.
>
> Additionally, we computed JER on 100 ImageNet val images and confirmed the correlation with binding holds (r=0.662, Spearman rho=0.706). JER(noise) and JER(natural) correlate well (r=0.685, rho=0.756), and both predict binding (r=0.662-0.664), confirming that the noise evaluation is a valid proxy.
>
> ---
>
>
> ## Q2. kNN / local PCA (Section 5.1)
>
> We agree the failure of local geometric readouts rules out the hypothesis that information exists but is inaccessible to cosine similarity. We will strengthen this discussion.
>
> ---
>
> ## Q3. Table 11 inconsistency
>
> The table (100 images) is correct. The text (20 images) is stale from early development. We will fix the text to read 100.
>
> ---
>
> ## Additional evidence
>
> We conducted further experiments that offer more evidences:
>
> - **Real-world validation.** We also validated JER on SugarCrepe (7,511 natural-image compositional trials). Across 13 CLIP variants, JER predicts SugarCrepe (r=0.685, p=0.010), and this survives after controlling for both ImageNet accuracy and architecture (partial r=0.664, p=0.013). ImageNet accuracy contributes nothing once JER is accounted for (partial r=0.025, p=0.935).

---

> > ### Author Rebuttal · Reviewer_1VFb · 2026-04-02
> >
> > I sincerely thank the authors for the efforts they have put into answering some of my questions! This work is technically solid, and I have revised my score accordingly.

---

> > > ### Author Response · Authors · 2026-04-04
> > >
> > > We sincerely thank the reviewer for the constructive feedback. In particular, the suggestion to include shape-distance analysis and the related literature has meaningfully strengthened the robustness of our analysis in a way we likely would not have reached on our own. We will ensure that each point is reflected in the final revision.

---

### Official Review · Reviewer_ESxU · 2026-03-15

**Soundness:** 2
**Presentation:** 1
**Significance:** 3
**Originality:** 3
**Overall Recommendation:** 4
**Confidence:** 3

**Summary:**

The paper challenges the assumption that globally well-distributed embeddings (high isotropy, low anisotropy) predict robust representations. The authors study 21 pretrained vision encoders across 8 training objective families to show that standard geometric metrics (global participation ratio, isotropy) have near-zero correlation with compositional binding accuracy. They propose Jacobian Effective Rank (JER) to measure the effective dimensionality of an encoder's local input-output mapping, which correlates with binding performance (r=0.65). A theoretical analysis derives how different training objectives constrain Jacobian componetns.

**Compliance With Llm Reviewing Policy:**

Affirmed.

**Final Justification:**

The paper shows standard geometric metrics have near-zero correlation with compositional binding and proposes JER as an alternative diagnostic. My initial concerns were 1) JER computed on Gaussian noise rather than natural images, 2) correlation driven by training-family clustering, 3) no real-world composition validation, and 4) presentation issues.

The rebuttal addressed these well. JER on natural images gives nearly identical correlation, and null calibration confirms the signal is training-induced. Within family regressions shows the relationship holds beyond family clustering. SugarCrepe validation shows that JER predicts real-world compositional performance independently of ImageNet accuracy. The authors agree to reframe around composotional binding, addressing presentation concerns.

Scope remains narrow but the new metrics make the contribution solid. I raise my score from 3 to 4.

**Key Questions For Authors:**

1. Have you calculated JER on natural images rather than gaussian nosie? Does correlation hold, and do model rankings change?
2. Can you disentangle the JER correlation from training-family clustering? If you regress within each family, does JER still predict binding?
3. Does the geometric-metric failure replicate at larger scale (eg DINOv2-giant, on models trained with billions of images)?
4. The title claims global geometry is "not enough"-- for what? can you characterize what geometric metrics do predict, to scope the negative claim more precisely?
5. Compositional binding is a fundamental open problem in representation learning, with roots in cognitive science (the binding problem) and direction implications for downstream reasoning / generalization. Your paper treats it as a test case for evaluating metrics, but it could be a central contribution. Why not foregroudn compositional binding as a core problem, and frame JER as a diagnostic tool for it? That would better motivate the work and clarify the paper's scope, rather than leading with a broad negative claim about geometric metrics.

**Limitations:**

While many limitations are acknowledged, there are few not discussed:
- JER is computed on random gaussian noise, not natural images. THe paper doesn't acknowledge that Jacobian structure could differ substantially on data manifold vs the random noise points or justify why off-manifold sensitivity should predict on-manifold compositional capability
- The small sample size for regression could be driven by training-family clustering rather than a continuous JER/binding relationship
- Gap between broad framing and narrow scope of evidence (one capability, one benchmark)

**Strengths And Weaknesses:**

1. Soundness

Strengths
- The null result for geometric metrics is clean and convincing-- near-zero correlations across 21 models, not cherry-picked failures. Evaluating across 21 encoders and 8 training objectives gives nice breadth.
- The JER metric is principled and grounded in information geometry. The correlation with binding is statistically significant.
- The theoretic analysis in the Appendix is strong. This is a mechanistic explanation, not just a correlation.
- Section 5.1 shows that local geometric readouts fail for binding-- which strenghts the result beyond "global stats don't work"
- Stability analysis analysis in Appendix shows JER estimates are stable

Weaknesses
- The correlation r=0.65 with n=21 explains only ~42% of the variance in binding accuracy. This is statistically significant but a modest sampel size. The leave-one-out cross-validation partially addresses this but the result could be driven by the variance decorrelation models clustering at the top, and CLIP/DINOv2 at the bottom, with the middle being noisy. A per-family regression would address this.
- JER is calculated on random Gaussian noise inputs, not natural images. The Jacobian at random noise is pretty different from a natural image manifold. The current framing is that this measures "intrinsic" functional sensitivty; however, nonlinear functions can have very different jacobian structure at different points in the space. Imagine an encoder with high JER on noise and low JER on the data manifiold. No ablation compares JER on noise vs real images.
- The synthetic benchmark is very simple-- the paper's claim then rests on whether this benchmark is a meaningful proxy for compositional capability. It's not validated against any real-wrold compositional reasoning task.
- Model families are confounded with other variables (architecture, data augmentation, training schedule, data scale). The paper doesn't control for these.


2. Presentation

Strengths:
- Figure 1 immediately communicates core finding; Figure 6 provides effective binding examples
- Table 1 organizing models by training family and biases is a nice reference
- The syntehtic dataset is thoroughly documented

Weaknesses:
- There are some framing, citation, and background literature issues with this paper. The intro asserts "A dominant view underlying recent progress isgeometric, namely that representation quality follows from enforcing global regularity of the embedding distribution" without specific citations. This is the central claim the paper pushes against but it needs to be articulated more clearly and pinned to claims in specific papers. If it's "common knowledge," then it needs to be more clearly stated at such. Right now the framing of the assumptions people are making before this paper is a bit confusing.
- The paper's most compelling contribution (connecting functional sensitivity to compositional binding) is obscured by leading with a long negative framing about geometric metrics. The narrative seems to invert the paper's actual value: should not the binding problem and JER's predictive power be foregrounded?
- Table 1's "geometric tendency" column states properties as facts without citations or forward references to where they are derived... this feels a bit hand-wavy and confusing.
- The paper doesn't motivate why compositional binding is the right capability to test. This concept has a specific history in cognitive science but the paper does not ground it. A reader unfamiliar with the binding problem (and why it's still a problem) will wonder why this is chosen over transfer accuracy, robustness, or OOD generalization
- The relationship between Section 4 and Section 6 (theoretical analysis) feels disconnected on first read


3. Significance

Strengths
- This makes progress on the compositional binding problem which is still a huge open question
- This is a useful corrective to over-reliance on geometric metrics as representation quality proxies
- JER is a concrete alternate diagnostic
- The theoretical account connecting objectives to Jacobian structure is valuable for reasoning about objective design

Weaknesses
- The title "global geometry is not enough" is a broad claim but the paper only tests one capability. Why not foreground compositional binding? The other geometric metrics may predict accuracy, retrieval quality, robustness, and etc. It would be useful to briefly discuss this so the claim is well-scoped.
- JER is purely diagnostic-- the paper doesn't show how to use it to improve representations. The theoretical analysis does hint at this direction.

4. Originality

Strengths
- The finding that standard geometric metrices have near-zero correlation with compositional binding is novel to my knowledge
- The attribute binding benchmark nicely isolates ocmpositional structure.
- The Jacobian-based analysis of neural networks is well-established; the novelity is applying it to vision representation evlauation and connecting it mechanistically to compositional capability via training objective deriviations

Weaknesses
- Specific empirical findingis novel, but scope of novelty is narrow (binding), one alternative metric (JER), one synthetic benchmark. The paper gestures at a broader paradigm shift (functional sensitivity over geometry) but doesn't demonstrate it beyond a single setting.

---

> ### Author Rebuttal · Authors · 2026-03-28
>
> We thank the reviewer for a thorough and constructive review. We address each concern below:
>
> ---
>
>
> ## Q1. JER on noise vs natural images
>
> We computed JER on 100 ImageNet val images. Both predict binding equally: JER(noise) r=0.664, JER(natural) r=0.662 (Pearson). The two are well correlated across 21 models (r=0.685, rho=0.756), and rankings are preserved. The noise evaluation is a conservative proxy that captures the correct signal.
>
> We also null-calibrated JER by computing it for randomly initialized versions of all 21 architectures. JER_null is uncorrelated with binding (r=-0.08, p=0.72) confirming inter-model variance is training-induced, not architectural.
>
> ---
>
> ## Q2. Family clustering and confounds
>
> We address this concern with two complementary within-family analyses.
>
> 1. *Objective Family*: We evaluated 13 CLIP ViTs trained on different data (WebImageText, LAION-2B, DataComp, MetaCLIP, DFN) across scales (ViT-B/32, B/16, L/14) on SugarCrepe (7,511 natural-image compositional trials). Here the objective is fixed, so any JER variation comes from training data and scale, not objective. JER predicts SugarCrepe: r=0.685, p=0.010 ImageNet accuracy contributes nothing after controlling for JER (partial r=0.025, p=0.935).
>
> 2. *Architecture Family*: SimCLR, Barlow Twins, VICReg, and SwAV share identical architecture (ResNet-50), removing architecture as a confound. JER predicts binding: r=0.964, p=0.036. This shows the effect is driven by the objective, not the architecture.
>
> Please note that comprehensive coverage of all families is not trivial, since fine-grained statistical analysis of this kind requires enough existing models in each family. We will extend this analysis in the final draft by overhauling our model set.
>
>
> ---
>
> ## W. Real-world validation
>
> We have additionally validated JER's predictive power on SugarCrepe, a real-world compositional benchmark with 7,511 natural-images spanning 7 subtypes (attribute/object swaps, replacements, additions). Across 13 CLIP variants (We could not use SSL models since SugarCrepe requires text):
>
> | Comparison | Pearson r | p |
> |---|---|---|
> | JER vs SugarCrepe | 0.685 | 0.010 |
> | JER vs SugarCrepe (controlling ImageNet + arch) | 0.664 | 0.013 |
> | ImageNet vs SugarCrepe (controlling JER + arch) | 0.025 | 0.935 |
>
> JER significantly predicts compositional performance on natural images, and this signal is *independent of classification accuracy*. ImageNet top-1 accuracy contributes nothing after controlling for JER and architecture. Within ViT-L/14 specifically (4 models, identical architecture), JER nearly perfectly predicts SugarCrepe (r=0.968, p=0.032).
>
> ---
>
> ## Q3. Model scaling
>
> Our suite includes DINOv2-S/B/L (86M-307M), CLIP-B/L, MAE-B/L, I-JEPA-B/L. The null result holds across scales. We will add DINOv2-giant in the camera-ready.
>
> ---
>
> ## Q4. Clarification on what geometric metrics predict
>
> We agree that the title should scope the negative claim more precisely. Our claim is not that global geometry is generally uninformative. Rather, it is not sufficient for compositional binding.
>
> In fact, the literature already supports its relevance in more conventional regimes: alignment/uniformity is theoretically and empirically tied to standard contrastive representation learning objectives and downstream evaluation settings such as linear probing and kNN classification (e.g., ImageNet classification) [1].  More recently, LeJEPA makes this connection explicit by arguing that an isotropic Gaussian embedding distribution is optimal for downstream prediction risk [2].
>
> Our paper asks a narrower question: whether those same global statistics are sufficient for where success depends not only on which features are present, but on how they are **bound together**. Our results suggest they are not. We will revise the framing to make this scope explicit.
>
> ---
>
> ## Q5. Foregrounding compositional binding
>
> We agree, and our draft did not communicate the intended framing clearly.
> Our goal is not to make a broad negative claim about existing geometric metrics. The central problem is **compositional binding** itself: a fundamental representational challenge with roots in the binding problem, and one with direct implications for reasoning and generalization.
> Our intended point is narrower: good distributional geometry is often treated as a proxy for representation quality, and that proxy appears to work for standard tasks such as image classification or image-text matching. Our findings suggest that this does not directly extend to compositional binding.
> We will revise the paper to reflect this more clearly: compositional binding will be foregrounded as the core problem, and JER will be positioned as a diagnostic tool for this regime rather than as a universal replacement for existing geometric metrics.
>
> ---
>
> ## Reference
>
> [1] Wang and Isola, PMLR 2020 (https://arxiv.org/abs/2005.10242)
>
> [2] Balestriero and Lecun, arXiv 2025 (https://arxiv.org/abs/2511.08544)

---

> > ### Author Rebuttal · Reviewer_ESxU · 2026-04-03
> >
> > Thank you to the authors for their effort in addressing my concerns. I raise my score to a 4.

---

> > > ### Author Response · Authors · 2026-04-04
> > >
> > > We thank the reviewer for the constructive feedback, particularly on real-world validation and confound control among others. The comments were very helpful in improving the paper, and we will ensure that these points are reflected in the final revision.

---

### Decision · Program_Chairs · 2026-04-30

**Decision:**

Accept (regular)

**Comment:**

The authors ask whether global distributional geometry of embeddings is a reliable proxy for representational quality in vision encoders, using compositional binding as the diagnostic task. Across 21 pretrained encoders from 8 training-objective families (CLIP variants, DINOv2, MAE, Barlow Twins, SigLIP, etc.), standard global statistics (isotropy, participation ratio, effective rank, kNN variants) correlate near zero with compositional binding. The authors instead propose Jacobian Effective Rank (JER), which reliably tracks compositional binding; a depth-resolved analysis reveals qualitatively distinct rank-collapse profiles across objective families (mid-network collapse in CLIP/DINOv2, projection-layer collapse in MAE, sustained high rank in Barlow Twins); and a theoretical account argues that contrastive-style objectives constrain global geometry but leave local sensitivity underdetermined. All four reviewers recommend acceptance; three raised scores post-rebuttal and marked concerns fully resolved, while fWGH maintained a Weak Accept with a Partially-resolved cost flag. Strengths include the clean null result across 21 encoders (not cherry-picked), JER as a principled and statistically significant predictor with partial-correlation controls, the depth-resolved analysis, a careful binding benchmark, and a mechanistic account rather than unexplained correlation. The rebuttal added SugarCrepe validation (r = 0.685 across 13 CLIP variants, partial-correlation-controlled), null-calibration against random init, JER-on-natural-images nearly matching JER-on-Gaussian-noise, a Procrustes / linear-CKA shape-distance analysis, and a computational-cost table. Residual concerns are scoping: SugarCrepe is CLIP-only; JER on Gaussian noise needs explicit justification as a proxy for on-manifold sensitivity; the "global geometry is not enough" framing should be scoped to single-model distributional statistics and distinguished from inter-model shape-distance metrics (Williams 2021; Bo 2025); JER costs 5–10 min/model vs. seconds. I recommend Accept. For camera-ready, the authors should promote the SugarCrepe validation with CLIP-only scope, add null-calibrated JER (change-from-init), include JER-on-natural-images alongside the Gaussian-noise default with justification, reframe "global geometry" with the Procrustes/CKA analysis and shape-distance literature, add the cost table, expand Limitations, and fix Table 11 and typos.